# When Diffusion Models Memorize: Inductive Biases in Probability Flow of Minimum-Norm Shallow Neural Nets

Chen Zeno [1]  Hila Manor [1]  Greg Ongie [2]  Nir Weinberger [1]  Tomer Michaeli [1]  Daniel Soudry [1]

## Abstract

While diffusion models generate high-quality images via probability flow, the theoretical understanding of this process remains incomplete. A key question is when probability flow converges to training samples or more general points on the data manifold. We analyze this by studying the probability flow of shallow ReLU neural network denoisers trained with minimal $\ell^2$ norm. For intuition, we introduce a simpler score flow and show that for orthogonal datasets, both flows follow similar trajectories, converging to a training point or a sum of training points. However, early stopping by the diffusion time scheduler allows probability flow to reach more general manifold points. This reflects the tendency of diffusion models to both memorize training samples and generate novel points that combine aspects of multiple samples, motivating our study of such behavior in simplified settings. We extend these results to obtuse simplex data and, through simulations in the orthogonal case, confirm that probability flow converges to a training point, a sum of training points, or a manifold point. Moreover, memorization decreases when the number of training samples grows, as fewer samples accumulate near training points.

## 1. Introduction

In diffusion models (Sohl-Dickstein et al., 2015; Ho et al., 2020; Song et al., 2021b), new images are sampled from the data distribution through an iterative process. Beginning with a random initialization, the model gradually denoises the image until a final image emerges. At their core, diffusion models learn the data distribution by estimating the score function of a Gaussian-blurred version of the data distribution. The connection between the score function and the denoiser, often called Tweedie's identity (Robbins, 1956; Miyasawa et al., 1961; Stein, 1981), holds only under optimal Bayes estimation. Moreover, for the estimated score to be a true gradient field, the denoiser must have a symmetric positive semidefinite Jacobian matrix (Chao et al., 2023; Manor & Michaeli, 2024). However, in practice, neural network denoisers are used, and their Jacobian matrix is generally non-symmetric, raising open questions about the convergence of the sampling process in score-based diffusion algorithms.

Diffusion models typically use a stochastic sampling process, which can be described by a stochastic differential equation (SDE) (Song et al., 2021b). Alternatively, a deterministic version of the sampling process can also be used, formulated as an ordinary differential equation (ODE) (Song et al., 2021a), called the probability flow ODE. We aim to theoretically analyze the probability flow, in order to illuminate this complex sampling process. However, practical diffusion architectures are typically deep and not fully connected, making it difficult to obtain theoretical guarantees without making additional strong assumptions (e.g., assuming a linearized regime, like the neural tangent kernel (Jacot et al., 2018)). Therefore, in this paper, we focus on diffusion models based on shallow ReLU neural network denoisers. Such networks are simple enough to allow for a theoretical investigation yet rich enough to offer valuable insights.

To gain insight into the dynamics of the probability flow ODE, we also explore a simpler ODE, which corresponds to flowing in the direction of the score of the noisy data distribution, for a fixed noise level. We call this the *score-flow* ODE. The score flow aims to sample from one of the modes of the noise-perturbed data distribution. We explore both the probability flow and the score flow ODEs for denoisers with minimal representation cost that perfectly fit the training data. Our analysis reveals that, for small noise levels, the trajectories of both flows are similar for a given initialization. However, the diffusion time scheduler induces "early stopping", which determines whether the probability flow converges to training samples or to other points on the data manifold. This analysis provides insights into the stability and convergence properties of these processes.

---

*Equal contribution  [1]Electrical and Computer Engineering, Technion, Haifa, Israel  [2]Mathematical and Statistical Sciences, Marquette University, Marquette, USA. Correspondence to: Chen Zeno <chenzeno@campus.technion.ac.il>.

*Proceedings of the $42^{nd}$ International Conference on Machine Learning*, Vancouver, Canada. PMLR 267, 2025. Copyright 2025 by the author(s).

**Our Contributions** We investigate the probability and the score flow of shallow ReLU neural network denoisers in the context of interpolating noisy samples with minimal cost, that is, when the denoiser maps each noisy sample exactly to its corresponding clean training point, resulting in zero empirical loss. Our focus is on the low-noise regime, where noisy samples are well clustered.

- **Theoretical**: We prove that when the clean training points are orthogonal to one another, the probability flow and score flow follow a similar trajectory for a given initialization point. However, while the score flow converges only to a training point or to a sum of training points, the probability flow can also converge to a point on the boundary of the hyperbox whose vertices are all partial sums of the training points. This happens due to "early stopping" induced by the diffusion time scheduler. We generalize this result to the case where the training points are the vertices of an obtuse simplex.

- **Experimental**: We train shallow denoisers that interpolate the training data with minimal representation cost on orthogonal datasets. We start by empirically demonstrating that the score flow ODE corresponding to a single such denoiser typically converges either to a sum of training points, which we call *virtual training points*, or to a general point on the boundary of the hyperbox (it converges to a training point only in rare occasions). We then show that the probability flow ODE, which uses a sequence of denoisers for varying noise levels, also converges to virtual points and to the boundary of the hyperbox, albeit at a somewhat lower frequency compared to the training points. Finally, we show that generalization improves as the number of clean data points increases.

## 2. Setup and Review of Previous Results

We study the denoising problem, where we observe a vector $\boldsymbol{y} \in \mathbb{R}^d$ that is a noisy observation of $\boldsymbol{x} \in \mathbb{R}^d$, i.e. $\mathbf{y} = \mathbf{x} + \boldsymbol{\epsilon}$, such that $\mathbf{x}$ and $\boldsymbol{\epsilon}$ are statistically independent and $\boldsymbol{\epsilon}$ is Gaussian noise with zero mean and covariance matrix $\sigma^2 \boldsymbol{I}$. The MSE loss of a denoiser $\boldsymbol{h}(\boldsymbol{y})$ is

$$\mathcal{L}_{\mathrm{MSE}}(\boldsymbol{h}) = \mathbb{E}_{\mathbf{x},\mathbf{y}} \|\boldsymbol{h}(\boldsymbol{y}) - \mathbf{x}\|^2, \qquad (1)$$

where the expectation is over the joint probability distribution of $\mathbf{x}$ and $\mathbf{y}$. The minimizer of the MSE loss is the MMSE estimator

$$\boldsymbol{h}_{\mathrm{MMSE}}(\boldsymbol{y}) = \mathbb{E}_{\mathbf{x}|\mathbf{y}} [\mathbf{x}|\mathbf{y} = \boldsymbol{y}]. \qquad (2)$$

In practice, since the true data distribution is unknown, we use empirical risk minimization with regularization. Consider a dataset consisting of $M$ noisy samples for each of the $N$ clean data points $\boldsymbol{x}_n$ such that $\boldsymbol{y}_{n,m} = \boldsymbol{x}_n + \boldsymbol{\epsilon}_{n,m}$, $n = 1, \ldots, N, m = 1, \ldots, M$. Then, one typically aims to minimize the loss

$$\mathcal{L}(\theta) = \frac{1}{MN} \sum_{m=1}^{M} \sum_{n=1}^{N} \|\boldsymbol{h}_\theta(\boldsymbol{y}_{n,m}) - \boldsymbol{x}_n\|^2 + \lambda C(\theta), \qquad (3)$$

where $\theta$ are the parameters of the denoiser model $\boldsymbol{h}_\theta$ and $C(\theta)$ is a regularization term. We focus on a shallow ReLU network with a skip connection as the parametric model of interest (Ongie et al., 2020; Zeno et al., 2023), given by

$$\boldsymbol{h}_\theta(\boldsymbol{y}) = \sum_{k=1}^{K} \boldsymbol{a}_k [\boldsymbol{w}_k^\top \boldsymbol{y} + b_k]_+ + \boldsymbol{V}\boldsymbol{y} + \boldsymbol{c}, \qquad (4)$$

where $\theta = (\{\theta_k\}_{k=1}^{K}; \boldsymbol{c}, \boldsymbol{V})$ with $\theta_k = (b_k, \boldsymbol{a}_k, \boldsymbol{w}_k) \in \mathbb{R} \times \mathbb{R}^d \times \mathbb{R}^d$ and $\boldsymbol{c} \in \mathbb{R}^d, \boldsymbol{V} \in \mathbb{R}^{d \times d}$ and the regularization term is a $\ell^2$ penalty on the weights, but not on the biases and skip connections, i.e.,

$$C(\theta) = \frac{1}{2} \sum_{k=1}^{K} \left( \|\boldsymbol{a}_k\|^2 + \|\boldsymbol{w}_k\|^2 \right). \qquad (5)$$

Zeno et al. (2023) showed that in the "low-noise regime", i.e. when the clusters of noisy samples around each clean data point are well-separated[1], there are multiple solutions minimizing the empirical MSE (first term in equation 3). Each of these solutions has a different generalization capability. They studied the solution at which the $\ell_2$ regularization of equation 5 is minimized.

**Definition 2.1.** Let $\boldsymbol{h}_\theta : \mathbb{R}^d \to \mathbb{R}^d$ denote a shallow ReLU network of the form of equation 4. For any function $\boldsymbol{h} : \mathbb{R}^d \to \mathbb{R}^d$ realizable as a shallow ReLU network, we define its **representation cost** as

$$R(\boldsymbol{h}) = \inf_{\theta: \boldsymbol{h} = \boldsymbol{h}_\theta} C(\theta)$$

$$= \inf_{\theta: \boldsymbol{h} = \boldsymbol{h}_\theta} \sum_{k=1}^{K} \|\boldsymbol{a}_k\| \ \text{ s.t. } \|\boldsymbol{w}_k\| = 1, \forall k, \qquad (6)$$

and a **minimizer** of this cost, i.e., a 'min-cost' solution, as

$$\boldsymbol{h}^* \in \operatorname*{argmin}_{\boldsymbol{h}} R(\boldsymbol{h}) \ \text{ s.t. } \boldsymbol{h}(\boldsymbol{y}_{n,m}) = \boldsymbol{x}_n \ \forall n, m. \qquad (7)$$

In the multivariate case, finding an exact min-cost solution for finitely many noise realizations is generally intractable. Therefore, Zeno et al. (2023) simplified equation 7 by assuming that $\boldsymbol{h}(\boldsymbol{y}) = \boldsymbol{x}_n$ for all $\boldsymbol{y}$ in an open ball centered at $\boldsymbol{x}_n$. Specifically, letting $B(\boldsymbol{x}_n, \rho)$ denote the ball of radius

---

[1]The noise level in the low-noise regime, though small, is not negligible and has been noted as practically "useful" (Zeno et al., 2023), e.g. for diffusion sampling (Raya & Ambrogioni, 2023).

$\rho$ centered at $\boldsymbol{x}_n$, we simplify notations by writing this constraint as $\boldsymbol{h}(B(\boldsymbol{x}_n, \rho)) = \{\boldsymbol{x}_n\}$. Consider minimizing the representation cost under this constraint, that is, solving

$$\boldsymbol{h}_\rho^*(\boldsymbol{y}) \in \underset{\boldsymbol{h}}{\arg\min}\, R(\boldsymbol{h}) \;\; \text{s.t.} \;\; \boldsymbol{h}(B(\boldsymbol{x}_n, \rho)) = \{\boldsymbol{x}_n\}, \;\; \forall n. \tag{8}$$

Even this surrogate problem is still challenging to solve explicitly in the general case. Nonetheless, it can be solved for two specific configurations of training data points, which serve as prototypes for more general configurations. The first case is when all the data points form an obtuse simplex, i.e., the generalization of an obtuse triangle to higher dimensions, and the second case is when the data points form an equilateral triangle (see Appendix D).

## 3. The Probability Flow and the Score Flow

Given an explicit solution for the neural network denoiser, we estimate the score function by leveraging the connection between the MMSE denoiser and the score function (Robbins, 1956; Miyasawa et al., 1961; Stein, 1981),

$$\boldsymbol{h}_{\text{MMSE}}(\boldsymbol{y}) = \boldsymbol{y} + \sigma^2 \nabla \log p(\boldsymbol{y}) , \tag{9}$$

where $p(\boldsymbol{y})$ is the probability density function of the noisy observation. From this relation, we can estimate the score function $\nabla \log p(\boldsymbol{y})$ as

$$\boldsymbol{s}(\boldsymbol{y}) = \frac{\boldsymbol{h}_\rho^*(\boldsymbol{y}) - \boldsymbol{y}}{\sigma^2} , \tag{10}$$

where $\boldsymbol{h}_\rho^*(\boldsymbol{y})$ is the minimum norm denoiser. In diffusion models, a stochastic process is typically used to sample new images. However, to generate unseen images from the data distribution, Song et al. (2021a) introduced a deterministic sampling process—the probability flow ODE (Song et al., 2021b; Karras et al., 2022).

We assume in this paper the variance exploding (VE) case, for which the probability flow ODE is given by

$$\forall t \in [0, T] : \frac{\mathrm{d}\boldsymbol{y}_t}{\mathrm{d}t} = -\frac{1}{2}\frac{\mathrm{d}\sigma_t^2}{\mathrm{d}t}\nabla \log p(\boldsymbol{y}_t, \sigma_t) , \tag{11}$$

where the score is estimated using the neural network denoiser $\nabla \log p(\boldsymbol{y}_t, \sigma_t) \approx \boldsymbol{s}(\boldsymbol{y}_t, \sigma_t)$, and $\sigma_t = \sqrt{t}$ is the diffusion time scheduler. The minus sign in the probability flow ODE arises due to the reverse time variable: we initialize at $\boldsymbol{y}_T$, and finish at $\boldsymbol{y}_0$, a sample from the data distribution. In Appendix A we show that by using time re-scaling arguments the probability flow ODE is equivalent to the following ODE

$$\frac{\mathrm{d}\boldsymbol{y}_r}{\mathrm{d}r} = \boldsymbol{h}_{\rho_{g_r^{-1}}}^*(\boldsymbol{y}_r) - \boldsymbol{y}_r, \tag{12}$$

where $g_r = -\log \sigma_r$, assuming the radius of the noise balls satisfies $\rho_t = \alpha \sigma_t$ for some $\alpha > 0$.

Additionally, we will also analyze the score flow, which is a simplified case of equation 12 where $\rho$ does not depend on $t$. Analyzing the score flow can be helpful in understanding the dynamics of the probability flow. The score flow represents the sampling process from one of the modes of the (multimodal) distribution of $\boldsymbol{y}$. The score flow is initialized at $\boldsymbol{y}_0$ and for $t > 0$ follows

$$\frac{\mathrm{d}\boldsymbol{y}_t}{\mathrm{d}t} = \nabla \log p(\boldsymbol{y}) . \tag{13}$$

Using the estimated score function and time re-scaling $r = \frac{1}{\sigma^2}t$ we obtain the score flow

$$\frac{\mathrm{d}\boldsymbol{y}_r}{\mathrm{d}r} = \boldsymbol{h}_\rho^*(\boldsymbol{y}_r) - \boldsymbol{y}_r . \tag{14}$$

Notably, in contrast to the probability flow ODE, the min-cost denoiser here is independent of $t$.

## 4. The Probability and Score Flow of Min-cost Denoisers

In this section, we consider training sets that model different types of data manifolds, and state for each type the possible convergence points of the score and probability flows of min-cost solutions. As the score flow is a specific instance of probability flow (after time re-scaling) in which the variance profile is fixed, the difference between the convergence points of these two flows thus illuminates the effect of the variance reduction scheduling $\sigma_t$ (and thus the $\rho_t$ schedule) on the generated sample.

We begin with the following simple, yet general, observation on the dynamics of score flow. For this dynamics, the stability condition for a stationary point $\boldsymbol{y}$ is that any eigenvalue of the Jacobian matrix of the score function with respect to the input $\boldsymbol{y}$, i.e., $\lambda(\boldsymbol{J}(\boldsymbol{y}))$ satisfies

$$\text{Re}\{\lambda(\boldsymbol{J}(\boldsymbol{y}))\} < 0 . \tag{15}$$

We next show that in any model that perfectly fits an open ball of radius $\rho > 0$ around the training points (and thus also interpolates the training set), the clean data points are stable stationary points of the score flow. This implies that, when initialized near these points, the process can converge to the clean data points.

**Proposition 4.1.** *Let $\rho > 0$ be arbitrary. Let $\boldsymbol{h}(\boldsymbol{y})$ be a denoiser that satisfies $\boldsymbol{h}(B(\boldsymbol{x}_n, \rho)) = \{\boldsymbol{x}_n\}$ for all $n \in [N]$ (and thus interpolates the training data). Then, any training point $\boldsymbol{y} \in \{\boldsymbol{x}_n\}_{n=1}^N$ is a stable stationary point of equation 13 where we estimate the score using $\boldsymbol{s}(\boldsymbol{y}) = \frac{\boldsymbol{h}(\boldsymbol{y}) - \boldsymbol{y}}{\sigma^2}$.*

*Proof.* For all $\boldsymbol{y} \in \{\boldsymbol{x}_n\}_{n=1}^N$ we get that $\boldsymbol{s}(\boldsymbol{y}) = 0$ since the denoiser interpolates the training data. In addition, for

all $\boldsymbol{y} \in \text{int}\,(B(\boldsymbol{x}_n, \rho)))$ the Jacobian matrix is

$$J\,(\boldsymbol{y}) = -\frac{1}{\sigma^2}\boldsymbol{I}\,, \tag{16}$$

therefore the stability condition of equation 15 holds. $\quad\square$

This result implies that, when the score function is differentiable and the training points are the only stationary points, the score flow will converge to the training points with probability 1.

### 4.1. Flow Properties on Analytically Solvable Datasets

Zeno et al. (2023) found the min-cost solution $\boldsymbol{h}_\rho^*$ analytically in three cases: (1) orthogonal points, (2) points that form an obtuse angle with one of the points, and (3) a specific case of 3 training points forming an equilateral triangle.

For simplicity, we consider the case of a dataset composed of orthogonal points, and defer to the appendices of other types of datasets. Specifically, suppose that we have $N$ training points $\{\boldsymbol{x}_n\}_{n=0}^{N-1}$ where $\boldsymbol{x}_0 = \boldsymbol{0}$ and the remaining training points are orthogonal, i.e., $\boldsymbol{x}_i^\top \boldsymbol{x}_j = 0$ for all $i, j > 0$ with $i \neq j$.[2] This approximates the behavior of data in many generic distributions (e.g., standard normal), which becomes more orthogonal in higher dimensions (Saxe et al., 2014; Boursier et al., 2022). For example, for standard i.i.d. Gaussian data $\boldsymbol{x}_n$, it can be shown using the analysis in Section 3.2.3 of Vershynin (2018) along with a union bound, that the largest cosine similarity between two distinct datapoints satisfies $\max_{n\neq m} \frac{\boldsymbol{x}_n \cdot \boldsymbol{x}_m}{|\boldsymbol{x}_n||\boldsymbol{x}_m|} \sim \sqrt{\frac{\ln N}{d}}$ with high probability. Therefore, in the realistic regime $\exp(d) \gg N > d \gg 1$, most pairs are nearly orthogonal. Let $\boldsymbol{u}_n = \boldsymbol{x}_n/\|\boldsymbol{x}_n\|$ for all $n = 1, ..., N-1$. A minimizer of equation 8, $\boldsymbol{h}_\rho^*$, is given by (Zeno et al., 2023, proof of Theorem 3)[3]

$$\boldsymbol{h}_\rho^*(\boldsymbol{y}) = \sum_{n=1}^{N-1} \left( \frac{\|\boldsymbol{x}_n\|}{\|\boldsymbol{x}_n\| - 2\rho} \Big( [\boldsymbol{u}_n^\top \boldsymbol{y} - \rho]_+ \right.$$
$$\left. - [\boldsymbol{u}_n^\top \boldsymbol{y} - (\|\boldsymbol{x}_n\| - \rho)]_+ \Big) \boldsymbol{u}_n \right). \tag{17}$$

We prove (Appendix D.1) the set of stationary points is the set of all possible sums of training points.

**Theorem 4.2.** *Suppose that the training points $\{\boldsymbol{x}_0, \boldsymbol{x}_1, \boldsymbol{x}_2, ..., \boldsymbol{x}_{N-1}\} \subset \mathbb{R}^d$ are orthogonal. Then, the set of the stable stationary points of equation 13 is $\mathcal{A} = \{\sum_{n \in \mathcal{I}} \boldsymbol{x}_n \mid \mathcal{I} \subseteq [N-1]\}$.*

---

[2]The result holds for the general case where $\boldsymbol{x}_0$ is non-zero, provided that $(\boldsymbol{x}_i - \boldsymbol{x}_0)^\top (\boldsymbol{x}_j - \boldsymbol{x}_0) = 0$.

[3]The same arguments used in the proof of Theorem 3 in (Zeno et al., 2023) can be applied to prove equation 17. The requirement for strictly obtuse angles (i.e., $\boldsymbol{x}_i^\top \boldsymbol{x}_j < 0$ instead of $\boldsymbol{x}_i^\top \boldsymbol{x}_j \leq 0$) in (Zeno et al., 2023) is only made specifically to ensure the uniqueness of the solution.

This implies that the stationary points are the vertices of a hyperbox (an $n$-dimensional analog of a rectangular). Next, we prove (in Appendix D.2) that the score flow converges to the vertex of the hyperbox closest to the initialization $\boldsymbol{y}_0$. Also, for some $\boldsymbol{y}_0$, score flow first converges to the hyperbox boundary, then to a specific vertex.

**Theorem 4.3.** *Suppose that the training points $\{\boldsymbol{x}_0, \boldsymbol{x}_1, \boldsymbol{x}_2, ..., \boldsymbol{x}_{N-1}\} \subset \mathbb{R}^d$ are orthogonal. Consider the score flow where we estimate the score using $\boldsymbol{s}\,(\boldsymbol{y}) = \frac{\boldsymbol{h}_\rho^*(\boldsymbol{y}) - \boldsymbol{y}}{\sigma^2}$ and an initialization point $\boldsymbol{y}_0$. If $\forall i \in [N-1] : \boldsymbol{u}_i^\top \boldsymbol{y}_0 \neq \frac{\|\boldsymbol{x}_i\|}{2}$, then*

- *We converge to the closest vertex of the hyperbox to the initialization $\boldsymbol{y}_0$.*

- *If the closest point to $\boldsymbol{y}_0$ on the hyperbox is a point on its boundary which is not a vertex, then for any $\epsilon < \min_i |\boldsymbol{u}_i^\top \boldsymbol{y}_0|$ there exists $\rho_0\,(\epsilon)$ and $T_0\,(\epsilon, \rho), T_1\,(\rho)$ such that for all $\rho < \rho_0\,(\epsilon)$ and all $T \in [T_0\,(\epsilon, \rho), T_1\,(\rho)]$, the point $\boldsymbol{y}_T$ is not a stable stationary point and at most at distance $\epsilon$ from the boundary of the hyperbox.*

Next, we consider the probability flow. For tractable analysis, we approximate the score estimator for small noise levels (i.e., for all $\min_{n \in [N-1]} \frac{\rho_t}{\|\boldsymbol{x}_n\|} \ll 1$) via Taylor's approximation to obtain (See Figure 16 in Appendix J for a comparison of the exact and approximated score function trajectories, which are nearly identical at low noise levels)

$$\boldsymbol{s}\,(\boldsymbol{y}, t) = \frac{1}{\sigma_t^2} \left( \sum_{n=1}^{N-1} \boldsymbol{u}_n \phi(\boldsymbol{u}_n^\top \boldsymbol{y}) - \left( \boldsymbol{I} - \sum_{n=1}^{N-1} \boldsymbol{u}_n \boldsymbol{u}_n^\top \right) \boldsymbol{y} \right) \tag{18}$$

where

$$\phi(z) = \begin{cases} -z & z < \rho_t \\ \rho_t \left( \frac{2}{\|\boldsymbol{x}_n\|} z - 1 \right) & \rho_t < z < \|\boldsymbol{x}_n\| - \rho_t \\ \|\boldsymbol{x}_n\| - z & z > \|\boldsymbol{x}_n\| - \rho_t \end{cases} \,. \tag{19}$$

With this approximation, one can show the probability flow and the score flow have a similar trajectory (for small $\rho$), if they have the same initialization point. However, the $\rho_t$ diffusion time scheduler in probability flow induces "early stopping". This can lead to the probability flow to converge to a non-vertex boundary point (in contrast to score flow), or to influence the speed of convergence to a stationary point. We show this in the following result for the probability flow (proved in Appendix D.3)

**Theorem 4.4.** *Suppose that the training points $\{\boldsymbol{x}_0, \boldsymbol{x}_1, \boldsymbol{x}_2, ..., \boldsymbol{x}_{N-1}\} \subset \mathbb{R}^d$ are orthogonal. Consider the probability flow where $\sigma_t = \sqrt{t}$, we estimate the score using equation 18, and $\boldsymbol{y}_T$ is the initialization point. If $\forall i \in [N-1] : \boldsymbol{u}_i^\top \boldsymbol{y}_T \neq \frac{\|\boldsymbol{x}_i\|}{2}$, then*

- *If the closest point to $\boldsymbol{y}_T$ on the hyperbox is a vertex, then we converge to this vertex.*

- *If the closest point to $\boldsymbol{y}_T$ on the hyperbox is not a vertex, then there exists $\tau(\boldsymbol{y}_T, \rho_T)$ such that we converge to the closest vertex to the initialization point $\boldsymbol{y}_T$ if $T > \tau(\boldsymbol{y}_T, \rho_T)$, and we converge to a point on the boundary of the hyperbox if $T < \tau(\boldsymbol{y}_T, \rho_T)$.*

Theorem 4.4 shows that the probability flow converges[4] to a vertex of the hyperbox or a point on the boundary of the hyperbox. We consider this hyperbox boundary as an implicit data manifold—the diffusion model samples from this hyperbox boundary even though we did not assume an explicit sampling model that generated the training data, such as a distribution supported on the manifold. However, in some cases probability flow ODE can converge to specific points in this manifold: the training points, or sums of training points ("virtual points"). We view these virtual points as representing a form of generalization, as they go beyond the empirical data distribution and form novel combinations not present in the training set.

This result aligns well with empirical findings that diffusion models can memorize individual training examples and generate them during sampling (Carlini et al., 2023). In addition, an empirical result shows that Stable Diffusion (Rombach et al., 2022) can reproduce training data by piecing together foreground and background objects that it has memorized (Somepalli et al., 2023). This behavior resembles our result that the probability flow can also converge to sums of training points. In Stable Diffusion we observe a "semantic sum" of training points; however, our analysis focuses on the probability flow of a simple 1-hidden-layer model, while in deep neural networks summations in deeper layers can translate into more intricate semantic combinations.

In Appendices B and C, we extend our results to non-orthogonal datasets. In Appendix B, we analyze training points forming an $(N - 1)$-simplex, where $\boldsymbol{x}_0$ forms an obtuse angle with all other vertices. We prove that the set of stable stationary points is a subset of all partial sums of the training points. Additionally, we show that when angles between data points are nearly orthogonal, a stable stationary point corresponding to the sum of all points exists. We then prove that the score flow first converges to a point along a chord connecting the origin to another training point before settling on an edge of the chord. Similarly, the probability flow converges either to a point on the chord or to one of its edges. In Appendix C, we consider training points forming an equilateral triangle. We show that the score flow first

---

[4]Note that, unlike the score flow, the probability flow ODE reaches the point exactly. Since this is a special case of convergence, and to maintain consistent terminology, we use the term "converge to a point" in both cases.

converges to the triangle's face before ultimately reaching a vertex.

## 5. Simulations

### 5.1. Probability Flow and Score Flow

In this section, we demonstrate the findings of Theorems 4.2, 4.3 and 4.4 in shallow neural networks. In practical settings, the continuous probability flow ODE given by equation 11 is discretized to $S$ timesteps, as

$$\boldsymbol{y}_{t-1} = \boldsymbol{y}_t + (\sigma_t^2 - \sigma_{t-1}^2)\frac{(\boldsymbol{h}_{\rho_t}^*(\boldsymbol{y}_t) - \boldsymbol{y}_t)}{2\sigma_t^2}, \quad t = T, \dots, 1, \tag{20}$$

where $\boldsymbol{h}_{\rho_t}^*(\boldsymbol{y}_t)$ is modeled as a series of $S$ denoisers (usually with weight sharing), which are applied consecutively to gradually denoise the signal. In this setting, the sampling should theoretically be initialized at $T = \infty$, however in practice it is initialized from a finite timestep $T$, which is chosen such that $\sigma_T \gg \|\boldsymbol{x}_i\|$ for all $i$. Similarly, the score-flow of equation 13 is discretized as

$$\boldsymbol{y}_{t+1} = \boldsymbol{y}_t + \gamma\frac{(\boldsymbol{h}_{\rho_{t_0}}^*(\boldsymbol{y}_t) - \boldsymbol{y}_t)}{\sigma_{t_0}^2}, \quad t = 0, 1, \dots, \tag{21}$$

where $\gamma$ is some step size and here $t_0$ is a fixed timestep (so that all iterations are with the same denoiser). Note that here $t$ increases along the iterations, and since we use a single denoiser, there is no constraint on the number of iterations we can perform.

It should be noted that while our theorems characterize only the low-noise regime, here we simulate a more practical sampling process, which starts the sampling from large noise. Namely, the initialization ($\boldsymbol{y}_T$ in equation 20 and $\boldsymbol{y}_0$ in equation 21) is drawn from a Gaussian with large $\sigma$. Thus, our theoretical analysis becomes relevant only once the dynamics enter the low-noise regime.

To demonstrate our results for the case of an orthogonal dataset, we use orthonormal training samples, set $\sigma_t = \sqrt{t}$, and choose $T = 100$ to ensure an effectively high noise at the beginning of the sampling process. We train a set of $S = 150$ denoisers, ensuring 50 equally-spaced noise levels in the "low-noise regime" and 100 equally-spaced noise levels outside it. Note that since we are discretizing an ODE, using sufficiently small timesteps ensures that the discretization does not affect the results. In practical models, some denoisers are guaranteed to operate in the low-noise regime, as the final sampled point is obtained when $t \approx 0$, where the noise level is inherently small. We train our networks on data in dimension $d = 30$, with $M = 500$ noisy samples per training sample, taking the dimension of the hidden layer of the networks to be $K = 300$.

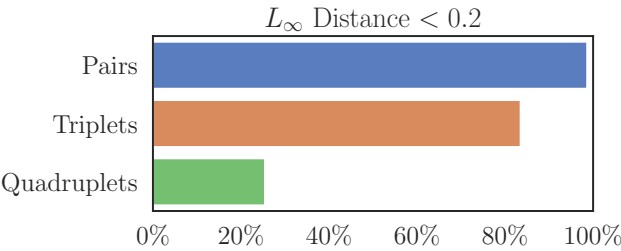

Figure 1: **Existence of stable virtual training points.** We run fixed-point iterations on a single denoiser, starting from all possible pair-wise, triplet-wise, and quadruplet-wise combinations of training samples. The plot shows the percentage of points that converged within an $L_\infty$ distance of 0.2 to the original, virtual, input point.

To be consistent with our theory, which assumes the denoiser achieves exact interpolation over the noisy training samples, we use a non-standard training protocol to enforce close-to-exact interpolation. Specifically, we pose the denoiser training as the equality constrained optimization problem

$$\min_\theta C(\theta) \ \text{s.t.} \ \boldsymbol{h}_\theta(\boldsymbol{y}_{n,m}) = \boldsymbol{x}_n, \ \forall n, m \qquad (22)$$

which we optimize using the Augmented Lagrangian (AL) method (see, e.g., (Nocedal & Wright, 2006)). Specifically, we define

$$\mathcal{L}_{AL}(\theta, \boldsymbol{Q}, \mu) := C(\theta) + \frac{1}{MN} \sum_{m=1}^{M} \sum_{n=1}^{N} \frac{\mu}{2} \|\boldsymbol{h}_\theta(\boldsymbol{y}_{n,m}) - \boldsymbol{x}_n\|^2$$
$$+ \langle \boldsymbol{q}^{(n,m)}, \boldsymbol{h}_\theta(\boldsymbol{y}_{n,m}) - \boldsymbol{x}_n \rangle \qquad (23)$$

where $\mu \in \mathbb{R}_{>0}$, $\boldsymbol{q}^{(n,m)} \in \mathbb{R}^d$ represents a vector of Lagrange multipliers, and $\boldsymbol{Q} \in \mathbb{R}^{d \times MN}$ is the matrix whose columns are $\boldsymbol{q}^{(n,m)}$ for all $m = 1, ..., M$, $n = 1, ..., N$. Then, starting from an initialization of $\mu_0 > 0$ and $\boldsymbol{Q}_0 = \boldsymbol{0}$, for $k = 0, 1, ..., \mathcal{K}$ we perform the iterative updates:

$$\theta_{k+1} = \arg\min_\theta \mathcal{L}_{AL}(\theta_k, \boldsymbol{Q}_k, \mu_k) \qquad (24)$$

$$\boldsymbol{q}_{k+1}^{(n,m)} = \boldsymbol{q}_k^{(n,m)} + \mu_k(\boldsymbol{h}_\theta(\boldsymbol{y}_{n,m}) - \boldsymbol{x}_n), \ \forall n, m \quad (25)$$

$$\mu_{k+1} = \eta \mu_k, \qquad (26)$$

where $\eta > 1$ is a fixed constant. The solution of equation 24 is approximated by following standard training using the Adam optimizer (Kingma & Ba, 2015) with a learning rate of $10^{-4}$ for $10^4$ iterations. We additionally take $\eta = 3$ and $\mathcal{K} = 7$, and decrease the learning rate by 0.5 after each iterative update.

We start by demonstrating the existence of *virtual* training points, that is, stable stationary points that are sums of training points, as predicted by Theorem 4.2. We take a denoiser from the "low-noise regime" ($\sigma_t = 0.095$ in this example)

and run 10 fixed-point iterations on all the predicted virtual points that consist of combinations of pairs, triplets and quadruplets of the training points. In Figure 1 we plot the percentage of these runs that converged within an $L_\infty$ distance of 0.2 to the predicted virtual point. As can be seen, 98.6% of the predicted virtual points composed of pairs of training points are stable in practice, and the stability of virtual points decreases as higher-numbers of combinations are considered. Nevertheless, the absolute number of stable virtual points increases substantially as higher-numbers of combinations are considered. Specifically, in the same example a total of 429, 3390, and 6965 stationary points were found for the pairs, triplet and quadruplet combinations. The increase in the absolute numbers is due to the higher number of higher-order sums. The decrease in percentages is due to small deviations in the ReLU boundaries of the trained denoiser compared to the theoretical optimal denoiser. These deviations have a greater impact on stationary points that involve sums of more training points.

Next, we explore the full dynamics of the diffusion process. We start with the score flow for a single denoiser from timestep $t_0$, which corresponds to noise level $\sigma_{t_0} = 0.095$. We randomly sample 500 points from $\mathcal{N}(0, 100\boldsymbol{I})$, and apply 3000 score-flow iterations to each, with a step size of $\gamma = 5 \cdot 10^{-4}$. The right hand side of Figure 2a shows the percentage of points that converged within an $L_\infty$ distance of 0.2 to either virtual points, training points, or a boundary of the hyperbox. On the left hand side of Figure 2a, we plot the projection of all samples on three dimensions. Out of 500 samples, almost all points converged to virtual points, which is expected in random initialization due to their larger number, compared to the training points. The rest of the points converged to the hyperbox's boundaries. The path the points take towards the hyperbox first draws them to the closest boundary, and then they drift along the boundary towards the closest stable stationary point.

Finally, we examine a full diffusion process with the probability ODE. Here we follow equation 20 using $S = 150$ trained denoisers, starting again from 500 randomly sampled points from $\mathcal{N}(0, \sigma_T \boldsymbol{I})$. Our results hold where the noise level is small compared to the norm of the training samples. Therefore, denoisers of large noise levels are not expected to have stable virtual points. In probability flow, most noise levels are large compared to this norm, as the sampling process begins with a large variance (in the VE case). Specifically, in our example only the last 50 denoisers have small noise levels. Yet, as can be seen on the right hand side of Figure 2b, a large percentage of the samples produced are virtual points. In contrast to the score flow case, the start of the sampling process here attracts most samples towards the mean of the training points, as any optimal-MSE denoiser would, which creates a biased starting point to the sampling process in the "low-noise

regime". From this regime onwards, the points travel along the boundaries of the hyperbox towards their nearest stable points, which is usually a training point. This behavior is demonstrated on the left side of Figure 2b, where the projected path of a random point is drawn starting from the $90^{\text{th}}$ step. See Appendix E for comparisons of additional thresholds and Appendix F for experiments where neural denoisers are trained using the standard Adam optimizer, with and without weight decay regularization. We show that without weight decay, the probability flow converges only to training points or boundary points of the hyperbox. In contrast, with weight decay, it also converges to virtual points, aligning with the results obtained using the Augmented Lagrangian method for training the minimum-norm denoiser. We conduct additional experiments on the effect of adding dropout to the training process. This results in a higher MSE loss, leading to fewer denoisers in the "low-noise regime" to perfectly fit the training data, and thus to more samples being generated outside the boundary of the hyperbox. Please see Appendix H for further details. Finally, Appendix I includes a comparison to detection metrics for memorization, spuriousness, and generalization as proposed by Pham et al. (2024), using the same setup as in Figure 2.

### 5.2. The Effect of the Number of Training Samples

The effect of the training set size has been explored in several past works (Somepalli et al., 2023; Kadkhodaie et al., 2024), as surveyed in detail in Section 6. Here we continue the analysis to investigate the effect of changing $N$, the training set size, on the full dynamics of the diffusion process with the probability ODE. Specifically, we repeat the experiment while reducing $N$. All the hyperparameters are kept the same, except for $M$ which we increase to 2000 for $N = 10$ only, to prevent over-fitting in the large-noise regime. Figure 3 shows the percentage of points that converged within an $L_\infty$ distance of 0.2 to either virtual points, training points, or a boundary of the hyperbox, for the different $N$ values. The generalization increases with $N$, drawing a larger percentage of samples to converge in the vicinity of virtual points, or to boundaries of the hyperbox. This aligns with the results of Kadkhodaie et al. (2024).

As in the case of strictly orthogonal data it is impossible to have $N > d + 1$, we consider here two additional cases to explore similar setups: oversampling duplications from the training set, and augmenting the training set with samples randomly drawn from the boundary of the hyperbox. When considering the effect of oversampling duplications, previous works observed that diffusion models tend to over-fit more to duplicate training points than to other training points (Somepalli et al., 2023). However, here we study the regime in which the model perfectly fits all the training points. In practice, duplicate training points would cause the neural network to fit them better, at the expense of the other

training points. Then, we expect our analysis to effectively hold, but only for the training points that are well-fitted and their associated virtual points. Therefore, this mirrors the case of decreasing $N$, and will cause more convergence to the duplicated training points and increase memorization. Next, we augment the original orthonormal dataset used in Figure 2 with additional random data points drawn from the boundary of the hyperbox. We then retrain the denoisers using the AL method for two values: $N = 40$ and $N = 50$. In these cases, the probability flow still almost exclusively converges to the hyperbox boundary. Further details appear in Appendix G.

## 6. Related Work

**Memorization and Generalization in Deep Generative Models** Several recent works have sought to explain the transition from memorization to generalization in deep generative models, both from a theoretical and empirical perspective. One early line of work in this vein studied memorization in over-parametrized (non-denoising) autoencoders (Radhakrishnan et al., 2019; 2020). This work shows that over-parameterized autoencoders trained to low cost are locally contractive about each training sample, such that training images can be recovered by iteratively applying the autoencoder to noisy inputs. A theoretical explanation of this phenomenon using a neural tangent kernel analysis is given in (Jiang & Pehlevan, 2020). More recent work has also shown that state-of-the-art diffusion models exhibit a similar form of memorization, such that extraction of training samples is possible by identifying stable stationary points of the diffusion process (Carlini et al., 2023). Additionally, when trained on few images, several works have shown that the outputs of diffusion models are strongly biased towards the training set, and thus fail to generalize (Somepalli et al., 2023; Yoon et al., 2023; Kadkhodaie et al., 2024). A recent empirical study suggests that memorization and generalization in diffusion models are mutually exclusive phenomenon, and successful generation occurs only when memorization fails (Yoon et al., 2023; Zhang et al., 2024). Beyond these empirical studies, recent work has put forward theoretical explanations for generalization in score-based models. In (Pidstrigach, 2022), the authors show that score-based models can learn manifold structure in the data generating distribution. A complementary perspective is provided by Kadkhodaie et al. (2024), which argues that diffusion models implicitly encode geometry-adaptive harmonic representations. Ross et al. (2025) propose the Manifold Memorization Hypothesis as a geometric framework for understanding memorization in deep generative models. Using local intrinsic dimension to compare the model's learned manifold with the data manifold, they distinguish between overfitting-driven and data-driven memorization.

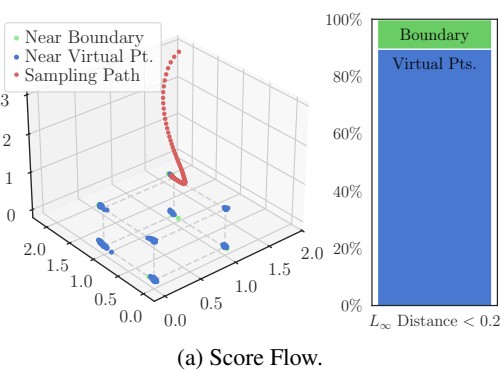

(a) Score Flow.

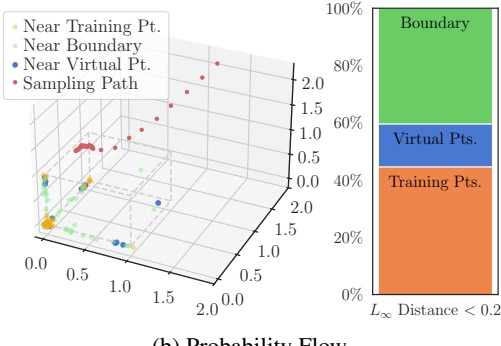

(b) Probability Flow.

Figure 2: **Projection to three dimensions and convergence types frequency of randomly sampled points.** We run the discrete ODE formulation of equation 20 for 500 randomly sampled points from $\mathbb{R}^{30}$, for both sampling using the score flow (2a) and a regular diffusion process (2b). For each, we plot on the right the percentage of points that converged to either a virtual point, a training point, or to the boundaries of the hyperbox, out of all points. On the left, we plot the sampling results projected to three dimensions, along with the path a single point took until convergence. In score flow, all points converged to either virtual points or to boundaries of the hyperbox, which is evident in the point clusters in the locations of the projected virtual points. For probability flow, the bias induced by the "large-noise regime" denoisers diffusion causes more samples to converge around the training points and their adjacent boundaries. Nevertheless, a large percentage of samples still converge in the vicinity of virtual points. The paths the points take towards the hyperbox draws them first to the closest boundary, and then, if the step sizes and amount permit, travel along the edges towards the closest stable stationary points.

**Representation Costs and Neural Network Denoisers**
Several other works have investigated overparameterized autoencoding/denoising networks with minimal representation cost (i.e., minimal $\ell^2$-norm of parameters). Function space characterizations of min-norm solutions of shallow fully connected neural networks are given in (Savarese et al., 2019; Ongie et al., 2020; Parhi & Nowak, 2021; Shenouda et al., 2024); extensions to deep networks and emergent bottleneck structure are considered in (Jacot, 2022; Jacot et al., 2022; Jacot, 2023; Wen & Jacot, 2024). The present work relies on the shallow min-norm solutions derived by Zeno et al. (2023) for specific configurations of data points, but goes beyond this work in studying the dynamics of its associated flows.

A recent study investigates properties of shallow min-norm solutions to a score matching objective (Zhang & Pilanci, 2024), building off of a line of work that studies min-norm solutions from a convex optimization perspective (Pilanci & Ergen, 2020; Ergen & Pilanci, 2020; Sahiner et al., 2021; Wang & Pilanci, 2021). In the case of univariate data, an explicit min-norm solution of the score-matching objective is derived, and convergence results are given for Langevin sampling with the neural network-learned score function. Additionally, in the multivariate case, general min-norm solutions to the score-matching loss are characterized as minimizers of a quadratic program. Our results differ from (Zhang & Pilanci, 2024) in that we study different optimization formulations (denoising loss versus score-matching loss) and inference procedures (probability- and score-flow versus Langevin dynamics). Our results focus on high-dimensional data belonging to a simplex, while Zhang & Pilanci (2024) give convergence guarantees only in the case of univariate data.

## 7. Discussion

**Conclusions.** We explored the probability flow ODE of shallow neural networks with minimal representation cost. We showed that for orthogonal dataset and obtuse-angle dataset the probability flow and the score flow follow the same trajectory given the same initialization point and small noise level. The diffusion time scheduler in probability flow induces "early stopping", which results in converging to a boundary point instead of a specific vertex (as in score flow) or speed up convergence to a specific vertex. Simulations confirm these findings and show that generalization improves as the number of clean training points increases. In practice, natural images often lie on a low-dimensional linear subspace due to their approximate low-rank structure (see Zeno et al. (2023), Appendix D3). As a result, the denoiser first contracts the data towards this subspace (see Theorem 2 in Zeno et al. (2023)). If the subspace has a sufficiently high dimension, then data points become approximately orthogonal. Thus, the geometric structure of high-dimensional image spaces may resemble the hyperbox studied in our work. One possible extension of this work is to analyze the probability flow ODE in the case of variance-preserving processes. This is an important case

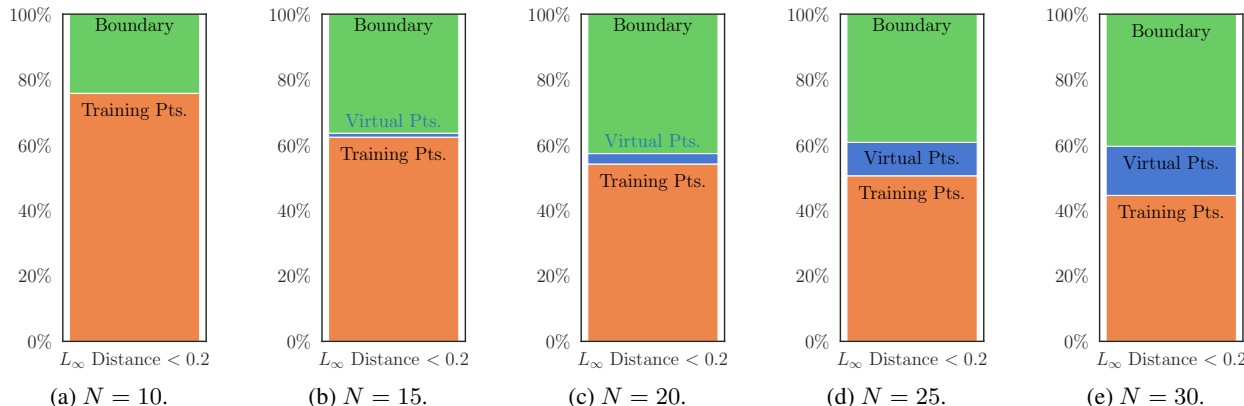

Figure 3: **Convergence types frequency of randomly sampled points in diffusion sampling for different $N$.** We run the discrete ODE formulation of equation 20 for 500 randomly sampled points from $\mathbb{R}^{30}$ for diffusion sampling, using different training set sizes, $N$. We plot the percentage of points that converged to either a virtual point, a training point, or to the boundaries of the hyperbox, out of all points. The generalization increases with $N$, drawing a larger percentage of samples to converge in the vicinity of virtual points and the boundaries of the hyperbox.

since practical diffusion models more often use variance-preserving forward and backward processes.

**Limitations.** A key limitation of our analysis is the assumption (inherited from Zeno et al. (2023)) that the denoiser interpolates data across a full $d$-dimensional ball centered around each clean training sample, where $d$ represents the input dimension. In real-world scenarios, the number of noisy samples is typically smaller than the input dimension $d$. A more accurate approach might involve assuming that the denoiser interpolates over an $(M-1)$-dimensional disc around each training sample, reflecting the norm concentration of Gaussian noise in high-dimensional spaces. Furthermore, for mathematical tractability, our analysis focuses on a single hidden layer model.

## Acknowledgements

The research of DS was Funded by the European Union (ERC, A-B-C-Deep, 101039436). Views and opinions expressed are however those of the author only and do not necessarily reflect those of the European Union or the European Research Council Executive Agency (ERCEA). Neither the European Union nor the granting authority can be held responsible for them. DS also acknowledges the support of the Schmidt Career Advancement Chair in AI. The research of NW was partially supported by the Israel Science Foundation (ISF), grant no. 1782/22. The research of TM was partially supported by the Israel Science Foundation (ISF), grant no. 2318/22. The research of GO was partially supported by the US National Science Foundation (NSF), grant no. 2153371.

## Impact Statement

This paper presents a theoretical analysis of diffusion models under specific constraints, aiming to enhance the understanding of generative models. This may lead to greater transparency when using these models. Moreover, we anticipate that insights gained from these simpler cases will shed light on the memorization and generalization behaviors in large-scale diffusion models, which pose privacy concerns. Lastly, we note the neural network examined in this paper is a shallow one, whereas practical contemporary implementations almost always involve deep networks. Naturally, addressing deep networks from the outset would pose an impassable barrier. In general, our guiding principle for research works that aim to understand new or not-yet-understood phenomena is that we should first study it in the simplest model that shows it, so as not to get distracted by possible confounders, and to enable a detailed analytic understanding. For example, when exploring or teaching a statistical problem issues, we would typically start with linear regression, understand the phenomena in this simple case, and then move on to more complex models. Thus, we hope the example we set in this paper will help promote this guiding principle for research and teaching.

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

## A. Proofs of Results in Section 3

The probability flow ODE is given by

$$\frac{\mathrm{d}\boldsymbol{y}_t}{\mathrm{d}t} = -\frac{1}{2}\frac{\mathrm{d}\sigma_t^2}{\mathrm{d}t}\nabla\log p\left(\boldsymbol{y}_t, \sigma_t\right) \tag{27}$$

$$= -\sigma_t \frac{\mathrm{d}\sigma_t}{\mathrm{d}t}\nabla\log p\left(\boldsymbol{y}_t, \sigma_t\right) . \tag{28}$$

First, we apply change of variable as follows

$$r = g\left(t\right) = -\log\sigma_t \tag{29}$$

$$\frac{\mathrm{d}r}{\mathrm{d}t} = -\frac{1}{\sigma_t}\frac{\mathrm{d}\sigma_t}{\mathrm{d}t} \tag{30}$$

$$\frac{\mathrm{d}t}{\mathrm{d}r} = \left(-\frac{1}{\sigma_t}\frac{\mathrm{d}\sigma_t}{\mathrm{d}t}\right)^{-1} . \tag{31}$$

Therefore,

$$\frac{\mathrm{d}y_t}{\mathrm{d}r} = \frac{\mathrm{d}y_t}{\mathrm{d}t}\frac{\mathrm{d}t}{\mathrm{d}r} = \left(-\sigma_t\frac{d\sigma_t}{dt}\nabla\log p\left(y_t, \sigma_t\right)\right)\left(-\frac{\sigma_t}{\frac{d\sigma_t}{dt}}\right) \tag{32}$$

$$= \sigma_t^2\nabla\log p\left(y_t, \sigma_t\right) \tag{33}$$

Next, we estimate the score function using a neural network denoiser, and substitute $t = g^{-1}\left(r\right)$ to obtain

$$\frac{\mathrm{d}y_r}{\mathrm{d}r} = h_{\rho(g^{-1}(r))}^*(y_r) - y_r . \tag{34}$$

## B. Obtuse-Angle Datasets

In this section we consider the case of a non-orthogonal dataset. Specifically, suppose the convex hull of the training points $\{\boldsymbol{x}_0, \boldsymbol{x}_1, ..., \boldsymbol{x}_{N-1}\} \subset \mathbb{R}^d$ is a $(N-1)$-simplex such that $\boldsymbol{x}_0$ forms an obtuse angle with all other vertices; we assume WLOG that $\boldsymbol{x}_0 = 0$. We refer to this as an obtuse simplex. Let $\boldsymbol{u}_n = \boldsymbol{x}_n/\|\boldsymbol{x}_n\|$ for all $n = 1, ..., N-1$. In this case, the minimizer $\boldsymbol{h}_\rho^*$ is still given by equation 17.

In Figure 4, we illustrate the normalized score flow for the case of an obtuse 2-simplex (see Figure 15 in Appendix J for the unnormalized score flow). The normalized score function is the score function multiplied by the log of the norm of the score and divided by the norm of the score. As shown, the training points are stationary points. Next, we prove (in Appendix D.4) that, in the general case of $N$ training points, the set of stable stationary points is a subset of the set of all partial sums of the training points. Additionally, we demonstrate that when the angles between data points are nearly orthogonal, a stable stationary point corresponding to the sum of the points exists.

**Theorem B.1.** *Suppose the convex hull of the training points $\{\boldsymbol{x}_0, \boldsymbol{x}_1, ..., \boldsymbol{x}_{N-1}\} \subset \mathbb{R}^d$ is an obtuse simplex. Then, the set $\mathcal{A}$ of the stable stationary points of equation 13 satisfies $\{\boldsymbol{x}_n\}_{n=0}^{N-1} \subseteq \mathcal{A} \subseteq \{\sum_{n\in\mathcal{I}}\boldsymbol{x}_n \mid \mathcal{I} \subseteq \{0, 1, \cdots, N-1\}\}$. In addition, the point $\sum_{n\in\mathcal{I}}\boldsymbol{x}_n$, where $\mathcal{I} \subseteq \{0, 1, \cdots, N-1\}$ and $|\mathcal{I}| \geq 2$ if $0 \notin \mathcal{I}$ and $|\mathcal{I}| \geq 3$ if $0 \in \mathcal{I}$, is a stable stationary point if $\min_{k\in\mathcal{I}}\left\{\sum_{i\in\mathcal{I}\backslash\{k\}}\boldsymbol{u}_k^\top\boldsymbol{u}_i\|\boldsymbol{x}_i\|\right\} > -\rho$.*

The condition $\min_{k\in\mathcal{I}}\sum_{i\in\mathcal{I}\backslash\{k\}}\boldsymbol{u}_k^\top\boldsymbol{u}_i\|\boldsymbol{x}_i\| > -\rho$ holds for almost orthogonal dataset (and $\rho > 0$).

Next, we prove (in Appendix D.5) that in the general case with $N$ training points, for small noise levels (i.e., small $\rho$) and an initialization point close to the chords connecting the origin to each training point ($\boldsymbol{x}_n$), the score flow first converges to a point along a chord connecting the origin and another training point, and then to an edge of the chord ($\boldsymbol{0}$ or $\boldsymbol{x}_n$, depending on initialization).

**Theorem B.2.** *Suppose the convex hull of the training points $\{\boldsymbol{x}_0, \boldsymbol{x}_2, ..., \boldsymbol{x}_{N-1}\} \subset \mathbb{R}^d$ is an obtuse simplex. Given an initial point $\boldsymbol{y}_0$ such that $\rho < \boldsymbol{u}_i^\top\boldsymbol{y}_0 < \|\boldsymbol{x}_i\| - \rho$ and $\boldsymbol{u}_j^\top\boldsymbol{y}_0 < \rho$ for all $j \neq i$, consider the score flow where we estimate the score using $\boldsymbol{s}\left(\boldsymbol{y}\right) = \frac{\boldsymbol{h}_\rho^*(\boldsymbol{y})-\boldsymbol{y}}{\sigma^2}$. Then we converge to the closest edge of the chord. In addition, for all $\epsilon \in (0, \boldsymbol{u}_i^\top\boldsymbol{y}_0)$ there exists $\rho_0\left(\epsilon\right)$ and $T_0(\epsilon, \rho), T_1(\rho)$ such that for all $\rho < \rho_0\left(\epsilon\right)$ the point $\boldsymbol{y}_T$ is not a stable stationary point and at most at distance $\epsilon$ from the line between $\boldsymbol{x}_1$ and $\boldsymbol{x}_i$ for $T_0(\epsilon, \rho) < T < T_1(\rho)$.*

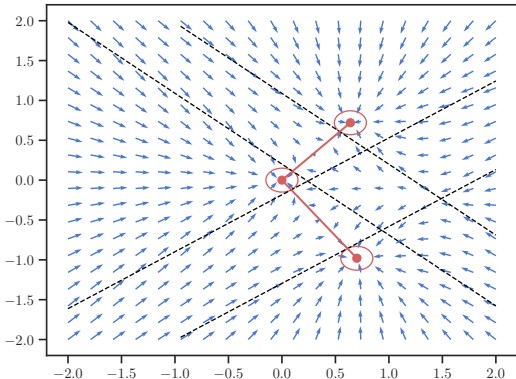

Figure 4: **The normalized score function of obtuse simplex**. The red dots are the training points $x_1, x_2, x_3$. The black lines are the ReLU boundaries.

We next turn to the probability flow. To this end, we assume that the initial point $y_T$ is such that $\rho_T < u_i^\top y_T < \|x_i\| - \rho_T$ and $u_j^\top y_T < \rho$ for all $j \neq i$. We again use Taylor's approximation in the small-noise level regime (specifically, for all $i \in [N-1] \frac{\rho_t}{\|x_n\|} \ll 1$), to obtain the following score estimation at a point $y$ such that $\rho_t < u_i^\top y < \|x_i\| - \rho_t$ and $u_j^\top y < \rho_t$ for all $j \neq i$ is

$$s(y, t) = \frac{1}{\sigma_t^2} \left( \left( \left( 1 + \frac{2}{\|x_i\|} \rho_t \right) u_i u_i^\top - I \right) y - \rho_t u_i \right). \tag{35}$$

We now have the following result regarding probability flow (proved in Appendix D.6)

**Theorem B.3.** *Suppose the convex hull of the training points $\{x_0, x_2, ..., x_{N-1}\} \subset \mathbb{R}^d$ is an obtuse simplex. Given an initial point $y_T$ such that $\rho_T < u_i^\top y_T < \|x_i\| - \rho_T$ and $u_j^\top y_T < \rho_T$ for all $j \neq i$. Consider the probability flow where $\sigma_t = \sqrt{t}$ and we estimate the score using equation 35. Then, $\exists \tau(y_T, \rho_T))$ such that we converge to a point on the line connecting $x_1$ and $x_i$ if $T < \tau(y_T, \rho_T)$ and if $T \geq \tau(y_T, \rho_T)$ we converge to the closest point in the set $\{x_0, x_i\}$ to $y_T$.*

Theorem B.3 shows that the probability flow converges to a point on the chord or to one of the edges of the chord. In this scenario, we consider the chords as the implicit data manifold.

## C. An Equilateral Triangle Dataset

In this section we consider the score flow in the case where the training points form the vertices of an equilateral triangle (as this is the last remaining dataset case for which the min-cost denoiser is analytically solvable (Zeno et al., 2023)). In Figure 5 we illustrate the normalized score flow for the case of an equilateral triangle dataset.

We prove (in Appendix D.7) that, given an initialization point near the edge of the triangle, the score flow first converges to the face of the triangle (the implicit data manifold here) and then to the vertex closest to the initialization point $y_0$.

**Proposition C.1.** *Suppose the convex hull of the training points $x_1, x_2, x_3 \in \mathbb{R}^d$ is an equilateral triangle. Given an initial point $y_0$ such that $i \in \{1, 2\} - \frac{\|x_i\|}{2} + \rho < u_i^\top y_0 < \|x_i\| - \rho$ and $u_3^\top y < -\frac{\|x_3\|}{2} + \rho$, consider the score flow where we estimate the score using $s(y) = \frac{h_\rho^*(y) - y}{\sigma^2}$. Then we converge to the closest vertex to the $y_0$. In addition, for all $0 < \epsilon < (u_1 + u_2)^\top y_0 - \frac{\|x\|}{2}$ there exists $\rho_0(\epsilon)$ and $T_0(\rho, \epsilon), T_1(\rho)$ such that for all $\rho < \rho_0(\epsilon)$ the point $y_T$ is not a stable stationary point and at most $\epsilon$ distance from the line between $x_1$ and $x_2$ for $T_0(\rho, \epsilon) < T < T_1(\rho)$.*

Without loss of generality, we can permute the training points indices $\{1, 2, 3\}$ in the above result. The probability flow for this case can be also analyzed, similarly to what we did in previous cases.

## D. Proofs of Results in Section 4

In this section we use the following Propositions from (Zeno et al., 2023).

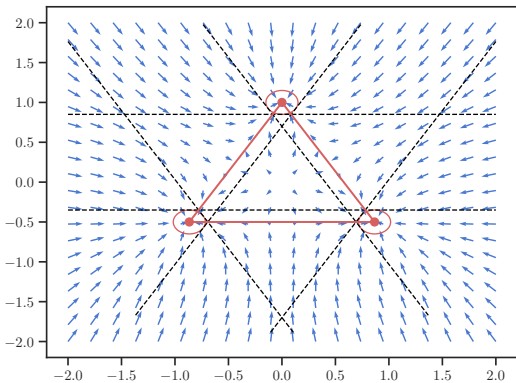

Figure 5: **The normalized score function of equilateral triangle**. The red dots are the training points $x_1, x_2, x_3$. The black lines are the ReLU boundaries.

**Proposition D.1.** *Suppose that the convex hull of the training points* $\{x_1, x_2, ..., x_N\} \subset \mathbb{R}^d$ *is a* $(N-1)$-*simplex such that* $x_1$ *forms an obtuse angle with all other vertices, i.e.,* $(x_j - x_1)^\top (x_i - x_1) < 0$ *for all* $i \neq j$ *with* $i, j > 1$. *Then the minimizer* $h_\rho^*$ *of equation 8 is unique and is given by*

$$h_\rho^*(y) = x_1 + \sum_{n=2}^{N} u_n \phi_n(u_n^\top (y - x_1)) \tag{36}$$

*where* $u_n = \frac{x_n - x_1}{\|x_n - x_1\|}$, $\phi_n(t) = s_n([t - a_n]_+ - [t - b_n]_+)$, *with* $a_n = \rho$, $b_n = \|x_n - x_1\| - \rho$, *and* $s_n = \|x_n - x_1\|/(b_n - a_n)$ *for all* $n = 2, ..., N$.

**Proposition D.2.** *Suppose the convex hull of the training points* $x_1, x_2, x_3 \in \mathbb{R}^d$ *is an equilateral triangle. Assume the norm-balls* $B_n := B(x_n, \rho)$ *centered at each training point have radius* $\rho < \|x_n - x_0\|/2$, $n = 1, 2, 3$, *where* $x_0 = \frac{1}{3}(x_1 + x_2 + x_3)$ *is the centroid of the triangle. Then a minimizer* $h_\rho^*$ *of equation 8 is given by*

$$h_\rho^*(y) = u_1 \phi_1(u_1^\top (y - x_0)) + u_2 \phi_2(u_2^\top (y - x_0)) + u_3 \phi_3(u_3^\top (y - x_0)) + x_0, \tag{37}$$

*where* $\phi_n(t) = s_n([t - a_n]_+ - [t - b_n]_+)$ *with* $u_n = \frac{x_n - x_0}{\|x_n - x_0\|}$, $a_n = -\frac{1}{2}\|x_n - x_0\| + \rho$, $b_n = \|x_n - x_0\| - \rho$, *and* $s_n = \|x_n - x_0\|/(b_n - a_n)$.

### D.1. Proof of Theorem 4.2

*Proof.* In the case of orthogonal dataset where for all $i \neq j$ $x_i^\top x_j = 0$ and $x_0 = 0$, the score function is

$$s(y) = \frac{h_\rho^*(y) - y}{\sigma^2} \tag{38}$$

$$= \frac{\sum_{i=1}^{N-1} e_n \frac{\|x_i\|}{\|x_i\| - 2\rho} ([y_i - \rho]_+ - [y_i - (\|x_i\| - \rho)]_+) - y}{\sigma^2}. \tag{39}$$

The Jacobian matrix is

$$J_{ij}(y) = \frac{\frac{\|x_i\|}{\|x_i\| - 2\rho} \Delta_i(y) \delta_{i,j} - \delta_{i,j}}{\sigma^2}, \tag{40}$$

where $\Delta_n(y)$ indicates if only one of the ReLU functions is activated. In matrix form, we obtain

$$J(y) = \frac{1}{\sigma^2} \left( \text{diag}\left( \frac{\|x_1\|}{\|x_1\| - 2\rho} \Delta_1(y), \cdots, \frac{\|x_{N-1}\|}{\|x_{N-1}\| - 2\rho} \Delta_{N-1}(y) \right) - I \right), \tag{41}$$

where $\Delta_n(y) \in \{0, 1\}$. In this case, the stability condition is

$$\text{Re}\{\lambda(J(y))\} = \lambda(J(y)) < 0. \tag{42}$$

Note that for $\Delta_i(\boldsymbol{y}) = 1$

$$\lambda(\boldsymbol{J}(\boldsymbol{y})) = \frac{\|\boldsymbol{x}_i\|}{\|\boldsymbol{x}_i\| - 2\rho} \Delta_i(\boldsymbol{y}) - 1 > 0. \tag{43}$$

Therefore, a stationary point is stable if and only if for all $i \in [N-1]$ $\Delta_i(\boldsymbol{y}) = 0$. We define the set $\mathcal{A} = \{\sum_{n \in \mathcal{I}} \boldsymbol{x}_n | \mathcal{I} \in \mathcal{P}([N-1])\}$. Note that the set of points where the score is zero and $\Delta_i(\boldsymbol{y}) = 0$ for all $i \in [N-1]$ is $\mathcal{A}$. $\qquad\square$

### D.2. Proof of Theorem 4.3

*Proof.* We assume WLOG that for all $i \in [N-1]$ $\boldsymbol{u}_i = \boldsymbol{e}_i$. We can analyze the ODE equation 14 along each orthogonal direction separately. In each direction, we divide the ODE into the following cases:

If $y_i \leq \rho$ or $i > N-1$, the score function is

$$s_i(y_i) = -\frac{y_i}{\sigma^2}. \tag{44}$$

Therefore, according to Lemma D.3,

$$(\boldsymbol{y}_t)_i = (\boldsymbol{y}_0)_i e^{-\frac{t}{\sigma^2}} \tag{45}$$

and we converge to zero.

If $y_i \geq \|\boldsymbol{x}_i\| - \rho$, the score function is

$$s_i(y_i) = \frac{\|\boldsymbol{x}_i\| - y_i}{\sigma^2}. \tag{46}$$

Therefore, according to Lemma D.3,

$$(\boldsymbol{y}_t)_i = (\boldsymbol{y}_0)_i e^{-\frac{t}{\sigma^2}} + \|\boldsymbol{x}_i\| \left(1 - e^{-\frac{t}{\sigma^2}}\right) \tag{47}$$

$$= (y_0 - \|\boldsymbol{x}_i\|) e^{-\frac{t}{\sigma^2}} + \|\boldsymbol{x}_i\| \tag{48}$$

and we converge to $\|\boldsymbol{x}_i\|$.

Finally, if $\rho < y_i < \|\boldsymbol{x}_i\| - \rho$, the score function is

$$s_i(y_i) = \frac{1}{\sigma^2} \left( \left( \frac{\|\boldsymbol{x}_i\|}{\|\boldsymbol{x}_i\| - 2\rho} - 1 \right) y_i - \frac{\|\boldsymbol{x}_i\| \rho}{\|\boldsymbol{x}_i\| - 2\rho} \right). \tag{49}$$

Therefore, according to Lemma D.3,

$$(\boldsymbol{y}_t)_i = (\boldsymbol{y}_0)_i e^{\left(\frac{\|\boldsymbol{x}_i\|}{\|\boldsymbol{x}_i\| - 2\rho} - 1\right)\frac{t}{\sigma^2}} + \frac{\|\boldsymbol{x}_i\|}{2} \left(1 - e^{\left(\frac{\|\boldsymbol{x}_i\|}{\|\boldsymbol{x}_i\| - 2\rho} - 1\right)\frac{t}{\sigma^2}}\right) \tag{50}$$

$$= \left((\boldsymbol{y}_0)_i - \frac{\|\boldsymbol{x}_i\|}{2}\right) e^{\left(\frac{\|\boldsymbol{x}_i\|}{\|\boldsymbol{x}_i\| - 2\rho} - 1\right)\frac{t}{\sigma^2}} + \frac{\|\boldsymbol{x}_i\|}{2}. \tag{51}$$

Here, if $(\boldsymbol{y}_0)_i = \frac{\|\boldsymbol{x}_i\|}{2}$ we converge to $\frac{\|\boldsymbol{x}_i\|}{2}$; if $(\boldsymbol{y}_0)_i > \frac{\|\boldsymbol{x}_i\|}{2}$ then we converge to $\|\boldsymbol{x}_i\|$; if $(\boldsymbol{y}_0)_i < \frac{\|\boldsymbol{x}_i\|}{2}$ then we converge to zero.

There are multiple initializations in which the closest point on the hyperbox is a point on the boundary which is not a vertex. We first consider the case where there exist a non empty set $\mathcal{I} \subset [N-1]$ such that for all $i \in \mathcal{I}$ $\rho < (\boldsymbol{y}_0)_i < \|\boldsymbol{x}_i\| - \rho$, and for all $j \in [N] \setminus \mathcal{I}$ $(\boldsymbol{y}_0)_j < \rho$ or $(\boldsymbol{y}_0)_j > \|\boldsymbol{x}_i\| - \rho$. We define $\Delta T_i(\rho)$ time to reach the edge of the partition, i.e. $\|\boldsymbol{x}_i\| - \rho$ (when $(\boldsymbol{y}_0)_i > \|\boldsymbol{x}_i\| - \rho$) starting from the initialization point, and $\Delta \tilde{T}_j(\rho, \epsilon)$ time to reach $\epsilon$ distance from zero or $\|\boldsymbol{x}_i\|$ starting from the initialization point:

$$\Delta T_i(\rho) = \sigma^2 \frac{\|\boldsymbol{x}_i\| - 2\rho}{2\rho} \log \left( \frac{\frac{\|\boldsymbol{x}_i\|}{2} - \rho}{(\boldsymbol{y}_0)_i - \frac{\|\boldsymbol{x}_i\|}{2}} \right) \tag{52}$$

$$\Delta \tilde{T}_j(\rho, \epsilon) = \sigma^2 \log \left( \frac{(\boldsymbol{y}_0)_i}{\epsilon} \right). \tag{53}$$

Since $\rho = \alpha\sigma$ we get that

$$\Delta T_i(\rho) = \rho \frac{\|\boldsymbol{x}_i\| - 2\rho}{2\alpha^2} \log\left(\frac{\frac{\|\boldsymbol{x}_i\|}{2} - \rho}{(\boldsymbol{y}_0)_i - \frac{\|\boldsymbol{x}_i\|}{2}}\right) \tag{54}$$

$$\Delta\tilde{T}_j(\rho, \epsilon) = \left(\frac{\rho}{\alpha}\right)^2 \log\left(\frac{(\boldsymbol{y}_0)_i}{\epsilon}\right). \tag{55}$$

Note that $\exists \rho_0(\epsilon) > 0$ such that $\forall \rho < \rho_0(\epsilon, )$

$$T_0 = \max_j \Delta\tilde{T}_j(\rho, \epsilon) < T < T_1 = \min_i \Delta T_i(\rho), \tag{56}$$

since $\exists \rho_0(\epsilon)$ such that

$$\left(\frac{\rho_0}{\alpha}\right)^2 \log\left(\frac{(\boldsymbol{y}_0)_i}{\epsilon}\right) < \rho_0 \frac{\|\boldsymbol{x}_i\| - 2\rho_0}{2\alpha^2} \log\left(\frac{\frac{\|\boldsymbol{x}_i\|}{2} - \rho_0}{(\boldsymbol{y}_0)_i - \frac{\|\boldsymbol{x}_i\|}{2}}\right) \tag{57}$$

$$\log\left(\frac{(\boldsymbol{y}_0)_i}{\epsilon}\right) < \frac{\|\boldsymbol{x}_i\| - 2\rho_0}{2\rho_0} \log\left(\frac{\frac{\|\boldsymbol{x}_i\|}{2} - \rho_0}{(\boldsymbol{y}_0)_i - \frac{\|\boldsymbol{x}_i\|}{2}}\right). \tag{58}$$

We can similarly derive the time interval during which $\boldsymbol{y}_T$ is at most $\epsilon$ distance from the boundary of the hyperbox and is not at a stationary point for additional initializations. Specifically, for all $i \in [N-1]$ $\rho < (\boldsymbol{y}_0)_i < \|\boldsymbol{x}_i\| - \rho$ is such an initialization point. $\qquad\square$

### D.3. Proof of Theorem 4.4

First, we prove the following lemma.

**Lemma D.3.** *consider the following affine ODE*

$$\frac{\mathrm{d}y_t}{\mathrm{d}t} = ay_t + b \tag{59}$$

*with initial point $y_T$, where $a \neq 0$. The solution is*

$$y = e^{a(t-T)}\left(y_T - \frac{b}{a}\left(e^{-a(t-T)} - 1\right)\right). \tag{60}$$

*Proof.* We verify directly that this is indeed the solution, since

$$\frac{\mathrm{d}y_t}{\mathrm{d}t} = ae^{a(t-T)}\left(y_T - \frac{b}{a}\left(e^{-at} - 1\right)\right) + e^{a(t-T)}be^{-a(t-T)} \tag{61}$$

$$= ae^{a(t-T)}\left(y_T - \frac{b}{a}\left(e^{-(t-T)t} - 1\right)\right) + b = ay_t + b \tag{62}$$

$$y_T = \left(y_T - \frac{b}{a}(1-1)\right) = y_T. \tag{63}$$

$\qquad\square$

Next, we prove the main Theorem.

*Proof.* We assume WLOG that for all $i \in [N-1]$ $\boldsymbol{u}_i = \boldsymbol{e}_i$. We can analyze the score flow along each orthogonal direction separately. In each direction, we divide the ODE to the following cases:

If $i \notin [N-1]$, then equation 12 is

$$\frac{\mathrm{d}y_r}{\mathrm{d}r} = -y. \tag{64}$$

Note that the initial point is at $r_0 = -\log \sqrt{T}$. Using Lemma D.3, we obtain

$$(\boldsymbol{y}_r)_i = (\boldsymbol{y}_T)_i \, e^{-1\left(r + \log \sqrt{T}\right)} . \tag{65}$$

Since $r = -\log \sqrt{t}$, we further obtain

$$(\boldsymbol{y}_t)_i = (\boldsymbol{y}_T)_i \, e^{\left(\log \sqrt{t} - \log \sqrt{T}\right)} = (\boldsymbol{y}_T)_i \, e^{\left(\log \sqrt{\frac{t}{T}}\right)} = (\boldsymbol{y}_T)_i \sqrt{\frac{t}{T}} . \tag{66}$$

Therefore, we obtain $(\boldsymbol{y}_0)_i = 0$.

We now consider now the case where $i \in [N-1]$.

In the case where $y_i < \rho_t$, equation 12 is

$$\frac{\mathrm{d}y_r}{\mathrm{d}r} = -y . \tag{67}$$

So, similarly to the previous case, we obtain $(\boldsymbol{y}_0)_i = 0$.

In the case where $y_i > \|x_i\| - \rho_t$, equation 12 is

$$\frac{\mathrm{d}y_r}{\mathrm{d}r} = \|\boldsymbol{x}_i\| - y . \tag{68}$$

Note that the initial point is at $r_0 = -\log \sqrt{T}$. Using Lemma D.3 we obtain

$$(\boldsymbol{y}_r)_i = e^{-1\left(r + \log \sqrt{T}\right)} \left((\boldsymbol{y}_T)_i + \|\boldsymbol{x}_i\| \left(e^{-1\left(r + \log \sqrt{T}\right)} - 1\right)\right) \tag{69}$$

$$= \|\boldsymbol{x}_i\| + ((\boldsymbol{y}_T)_i - \|\boldsymbol{x}_i\|) \, e^{-1\left(r + \log \sqrt{T}\right)} . \tag{70}$$

Since $r = -\log \sqrt{t}$, we further obtain

$$(\boldsymbol{y}_t)_i = \|\boldsymbol{x}_i\| + ((\boldsymbol{y}_T)_i - \|\boldsymbol{x}_i\|) \, e^{\left(\log \sqrt{t} - \log \sqrt{T}\right)} = \tag{71}$$

$$= \|\boldsymbol{x}_i\| + ((\boldsymbol{y}_T)_i - \|\boldsymbol{x}_i\|) \sqrt{\frac{t}{T}} . \tag{72}$$

Therefore, we obtain $(\boldsymbol{y}_0)_i = \|\boldsymbol{x}_i\|$.

In the case where $\rho_t < y_i < \|\boldsymbol{x}_i\| - \rho_t$, equation 12 is

$$\frac{\mathrm{d}y_r}{\mathrm{d}r} = \rho_{g_r^{-1}} \left(\frac{2}{\|\boldsymbol{x}_i\|} y - 1\right) . \tag{73}$$

Note that

$$\rho_t = \alpha \sigma_t = \alpha \sqrt{t} \tag{74}$$

$$g_r^{-1} = e^{-2r} . \tag{75}$$

Therefore,

$$\rho_r = \alpha e^{-r} \tag{76}$$

so we obtain the following ODE:

$$\frac{\mathrm{d}y_r}{\mathrm{d}r} = \alpha e^{-r} \left(\frac{2}{\|\boldsymbol{x}_i\|} y - 1\right) . \tag{77}$$

Next, we apply additional time re-scaling

$$k = -\alpha e^{-r} \tag{78}$$

$$\frac{\mathrm{d}k}{\mathrm{d}r} = \alpha e^{-r} = \rho_r \tag{79}$$

$$\frac{\mathrm{d}r}{\mathrm{d}k} = \alpha^{-1} e^r = \rho_r^{-1} . \tag{80}$$

So, we get the following ODE:

$$\frac{\mathrm{d}y_r}{\mathrm{d}k} = \frac{\mathrm{d}y_r}{\mathrm{d}r}\frac{\mathrm{d}r}{\mathrm{d}k} = \alpha e^{-r}\left(\frac{2}{\|\boldsymbol{x}_i\|}y - 1\right)\alpha^{-1}e^r = \frac{2}{\|\boldsymbol{x}_i\|}y - 1 \tag{81}$$

$$\frac{\mathrm{d}y_k}{\mathrm{d}k} = \frac{2}{\|\boldsymbol{x}_i\|}y - 1 . \tag{82}$$

Note that the initial point is at $k_0 = -\alpha\sqrt{T}$. Using Lemma D.3 we obtain

$$(\boldsymbol{y}_k)_i = e^{\frac{2}{\|\boldsymbol{x}_i\|}\left(k+\alpha\sqrt{T}\right)}\left((\boldsymbol{y}_T)_i + \frac{\|\boldsymbol{x}_i\|}{2}\left(e^{-\frac{2}{\|\boldsymbol{x}_i\|}\left(k+\alpha\sqrt{T}\right)} - 1\right)\right) \tag{83}$$

$$= \frac{\|x_i\|}{2} + \left((\boldsymbol{y}_T)_i - \frac{\|x_i\|}{2}\right)e^{\frac{2}{\|x_i\|}\left(k+\alpha\sqrt{T}\right)} . \tag{84}$$

Since $k = -\alpha e^{-r}$ and $r = -\log\sqrt{t}$, we obtain

$$(\boldsymbol{y}_r)_i = \frac{\|x_i\|}{2} + \left((\boldsymbol{y}_T)_i - \frac{\|x_i\|}{2}\right)e^{\frac{2}{\|x_i\|}\left(-\alpha e^{-r}+\alpha\sqrt{T}\right)} \tag{85}$$

$$(\boldsymbol{y}_t)_i = \frac{\|x_i\|}{2} + \left((\boldsymbol{y}_T)_i - \frac{\|x_i\|}{2}\right)e^{\frac{2}{\|x_i\|}\left(-\alpha\sqrt{t}+\alpha\sqrt{T}\right)} . \tag{86}$$

So, we obtain $(\boldsymbol{y}_0)_i = \frac{\|x_i\|}{2} + \left((\boldsymbol{y}_T)_i - \frac{\|x_i\|}{2}\right)e^{\frac{2\alpha\sqrt{T}}{\|x_i\|}}$. Given an initialization point $\boldsymbol{y}_T$, let $\mathcal{I} \subseteq [N-1]$ be a non empty set such that $\rho < (\boldsymbol{y}_T)_i < \|\boldsymbol{x}_i\| - \rho$ for all $i \in \mathcal{I}$ and either $(\boldsymbol{y}_T)_i < \rho$ or $(\boldsymbol{y}_T)_i > \|\boldsymbol{x}_i\| - \rho$ for all $j \in [N-1] \setminus \mathcal{I}$. Then, if

$$T > \max_{i\in\mathcal{I}}\left(\frac{\|x_i\|}{2\alpha}\right)^2\log^2\left(\frac{\frac{\|x_i\|}{2}}{(\boldsymbol{y}_T)_i - \frac{\|x_i\|}{2}}\right) , \tag{87}$$

we converge to the closest point in the set $\mathcal{A} = \{\sum_{n\in\mathcal{I}}\boldsymbol{x}_n \mid \mathcal{I} \subseteq [N-1]\}$ to the initialization point $\boldsymbol{y}_T$, where $\{\boldsymbol{x}_n\}_{n=0}^{N-1}$ is the training set. We instead converge to the closest boundary of the hyperbox to the initialization point $\boldsymbol{y}_T$ if

$$T < \max_{i\in\mathcal{I}}\left(\frac{\|x_i\|}{2\alpha}\right)^2\log^2\left(\frac{\frac{\|x_i\|}{2}}{(\boldsymbol{y}_T)_i - \frac{\|x_i\|}{2}}\right) . \tag{88}$$

$\square$

### D.4. Proof of Theorem B.1

*Proof.* In the case where the convex hull of the training points is an $(N-1)$-simplex, such that $\boldsymbol{x}_0$ forms an obtuse angle with all other vertices and $\boldsymbol{x}_0 = 0$, the score function is

$$\boldsymbol{s}(\boldsymbol{y}) = \frac{\boldsymbol{h}_\rho^*(\boldsymbol{y}) - \boldsymbol{y}}{\sigma^2} \tag{89}$$

$$= \frac{\sum_{n=1}^{N-1}\frac{\|\boldsymbol{x}_n\|}{\|\boldsymbol{x}_n\|-2\rho}\boldsymbol{u}_n\left([\boldsymbol{u}_n^\top\boldsymbol{y} - \rho]_+ - [\boldsymbol{u}_n^\top\boldsymbol{y} - (\|\boldsymbol{x}_n\| - \rho)]_+\right) - \boldsymbol{y}}{\sigma^2} . \tag{90}$$

The Jacobian matrix is

$$J_{ij}(\boldsymbol{y}) = \frac{\sum_{n=1}^{N-1}\frac{\|\boldsymbol{x}_n\|}{\|\boldsymbol{x}_n\|-2\rho}(u_n)_i(u_n)_j\Delta_n(\boldsymbol{y}) - \delta_{i,j}}{\sigma^2} , \tag{91}$$

where $\Delta_n(\boldsymbol{y})$ indicates if only one of the ReLU functions is activated. In matrix form we obtain

$$\boldsymbol{J}(\boldsymbol{y}) = \frac{1}{\sigma^2}\left(\boldsymbol{U}\boldsymbol{U}^\top - \boldsymbol{I}\right), \tag{92}$$

where

$$\boldsymbol{U} = \left(\Delta_1(\boldsymbol{y})\sqrt{\gamma_1}\boldsymbol{u}_1, \cdots, \Delta_{N-1}(\boldsymbol{y})\sqrt{\gamma_{N-1}}\boldsymbol{u}_{N-1}\right) \tag{93}$$

$$\gamma_n = \frac{\|\boldsymbol{x}_n\|}{\|\boldsymbol{x}_n\| - 2\rho} \tag{94}$$

$$\Delta_n(\boldsymbol{y}) \in \{0, 1\}. \tag{95}$$

Note that the Jacobian matrix is a real and symmetric matrix therefore it has real eigenvalues. In this case, the stability condition is

$$\mathrm{Re}\{\lambda(\boldsymbol{J}(\boldsymbol{y}))\} = \lambda(\boldsymbol{J}(\boldsymbol{y})) < 0. \tag{96}$$

For any $\boldsymbol{a} \in \mathbb{R}^d$

$$\boldsymbol{a}^\top \boldsymbol{J}(\boldsymbol{y})\boldsymbol{a} \le \lambda_{\max}(\boldsymbol{J}(\boldsymbol{y}))\boldsymbol{a}^\top \boldsymbol{a}. \tag{97}$$

This holds in particular for $\boldsymbol{a} \in \mathcal{S}^{d-1}$, therefore

$$\lambda_{\max}(\boldsymbol{J}) \ge \boldsymbol{a}^\top \frac{1}{\sigma^2}\left(\boldsymbol{U}\boldsymbol{U}^\top - \boldsymbol{I}\right)\boldsymbol{a} \tag{98}$$

$$= \frac{1}{\sigma^2}\left(\|\boldsymbol{a}^\top \boldsymbol{U}\|_2^2 - 1\right). \tag{99}$$

If we choose $\boldsymbol{a} = \boldsymbol{u}_n$ such that $\Delta_n(\boldsymbol{y}) \ne 0$, then $\|\boldsymbol{a}^\top \boldsymbol{U}\|_2^2 > 1$, since $\gamma_n > 1$. Therefore, a stationary point is stable if and only if for all $n \in \{1, \cdots, N-1\}$ $\Delta_i(\boldsymbol{y}) = 0$. Note that if $\boldsymbol{y}$ is such that $\Delta_n(\boldsymbol{y}) = 0$ for all $n \in \{1, \cdots, N-1\}$, then there exists $\mathcal{I} \in \mathcal{P}(0, 1, \cdots, N-1)$ such that

$$\boldsymbol{f}^*(\boldsymbol{y}) = \sum_{n \in \mathcal{I}} \boldsymbol{x}_n. \tag{100}$$

Therefore, $\boldsymbol{y}^* = \sum_{n \in \mathcal{I}} \boldsymbol{x}_n$ is a stationary point if and only if for all $i \in \{1, \cdots, N-1\}$ $\Delta_i(\boldsymbol{y}^*) = 0$. Note that the set of stable stationary points is not empty, since for all $i \in [N]$ the point $\boldsymbol{y}^* = \boldsymbol{x}_i$ is a stable stationary point because $\boldsymbol{f}^*(\boldsymbol{y}^*) = \boldsymbol{x}_i$, and thus $\Delta_n(\boldsymbol{y}^*) = 0$ for all $n \in \{1, \cdots, N-1\}$.

The condition for the point $\sum_{n \in \mathcal{I}} \boldsymbol{x}_n$ where $\mathcal{I} \subseteq [N]$ and $|\mathcal{I}| \ge 2$ if $0 \notin \mathcal{I}$ and $|\mathcal{I}| \ge 3$ if $0 \in \mathcal{I}$ to be a stable stationary point, is that for all $\forall k \in \mathcal{I}$

$$\sum_{i \in \mathcal{I}} \boldsymbol{u}_k^\top \boldsymbol{x}_i > \|\boldsymbol{x}_k\| - \rho, \tag{101}$$

which is equivalent to that for all $\forall k \in \mathcal{I}$

$$\sum_{i \in \mathcal{I}\backslash\{k\}} \boldsymbol{u}_k^\top \boldsymbol{x}_i > -\rho. \tag{102}$$

This set of inequality is equivalent to the condition

$$\min_{k \in \mathcal{I}}\left\{\sum_{i \in \mathcal{I}\backslash\{k\}} \boldsymbol{u}_k^\top \boldsymbol{u}_i \|\boldsymbol{x}_i\|\right\} > -\rho. \tag{103}$$

$\square$

## D.5. Proof of Theorem B.2

First, we prove the following lemma.

**Lemma D.4.** *Consider the following system of affine ODE*

$$\frac{\mathrm{d}\boldsymbol{y}_t}{\mathrm{d}t} = \boldsymbol{A}\boldsymbol{y}_t + \boldsymbol{b}\,, \tag{104}$$

*with the initial condition $\boldsymbol{y}_0$, where $\boldsymbol{A} \in \mathbb{R}^{d \times d}$ is a non singular matrix. The solution is*

$$\boldsymbol{y}_t = e^{\boldsymbol{A}t}\left(\boldsymbol{y}_0 - \boldsymbol{A}^{-1}\left(e^{-\boldsymbol{A}t} - \boldsymbol{I}\right)\boldsymbol{b}\right)\,. \tag{105}$$

*In the case where $\boldsymbol{A}$ is also symmetric, the solution can be written as*

$$\boldsymbol{y}_t = \sum_{i=1}^{d}\boldsymbol{v}_i\left(\boldsymbol{v}_i^\top \boldsymbol{y}_0\right)e^{\lambda_i t} - \sum_{i=1}^{d}\boldsymbol{v}_i\left(\boldsymbol{v}_i^\top \boldsymbol{b}\right)\lambda_i^{-1}\left(1 - e^{\lambda_i t}\right)\,, \tag{106}$$

*where $\sum_{i=1}^{d}\lambda_i\boldsymbol{v}_i\boldsymbol{v}_i^\top$ is the eigenvalue decomposition of the matrix $\boldsymbol{A}$.*

*Proof.* We verify directly that this is indeed the solution, since

$$\frac{\mathrm{d}\boldsymbol{y}_t}{\mathrm{d}t} = \boldsymbol{A}e^{\boldsymbol{A}t}\left(\boldsymbol{y}_0 - \boldsymbol{A}^{-1}\left(e^{-\boldsymbol{A}t} - \boldsymbol{I}\right)\boldsymbol{b}\right) + e^{\boldsymbol{A}t}e^{-\boldsymbol{A}t}\boldsymbol{b} = \boldsymbol{A}\boldsymbol{y}_t + \boldsymbol{b} \tag{107}$$

$$\boldsymbol{y}_0 = I\left(\boldsymbol{y}_0 - \boldsymbol{A}^{-1}\left(\boldsymbol{I} - \boldsymbol{I}\right)\boldsymbol{b}\right) = \boldsymbol{y}_0\,. \tag{108}$$

In the case where $\boldsymbol{A}$ is also symmetric,

$$e^{\boldsymbol{A}t} = \sum_{k=0}^{\infty}\frac{1}{k!}\left(\boldsymbol{A}t\right)^k = \boldsymbol{V}\left(\sum_{k=0}^{\infty}\frac{1}{k!}t^k\boldsymbol{\Lambda}^k\right)\boldsymbol{V}^\top \tag{109}$$

$$= \boldsymbol{V}\mathrm{diag}\left(e^{\lambda_1 t},\cdots,e^{\lambda_d t}\right)\boldsymbol{V}^\top = \sum_{i=1}^{d}e^{\lambda_i t}\boldsymbol{v}_i\boldsymbol{v}_i^\top \tag{110}$$

$$e^{-\boldsymbol{A}t} = \sum_{i=1}^{d}e^{-\lambda_i t}\boldsymbol{v}_i\boldsymbol{v}_i^\top\,. \tag{111}$$

Therefore,

$$\boldsymbol{y}_t = e^{\boldsymbol{A}t}\left(\boldsymbol{y}_0 - \boldsymbol{A}^{-1}\left(e^{-\boldsymbol{A}t} - \boldsymbol{I}\right)\boldsymbol{b}\right) \tag{112}$$

$$= \sum_{i=1}^{d}\boldsymbol{v}_i\boldsymbol{v}_i^\top e^{\lambda_i t}\left(\boldsymbol{y}_0 - \sum_{k=1}^{d}\boldsymbol{v}_k\boldsymbol{v}_k^\top\lambda_i^{-1}\sum_{j=1}^{d}\boldsymbol{v}_j\boldsymbol{v}_j^\top\left(e^{-\lambda_j t} - 1\right)\boldsymbol{b}\right) \tag{113}$$

$$= \sum_{i=1}^{d}\boldsymbol{v}_i\boldsymbol{v}_i^\top e^{\lambda_i t}\left(\boldsymbol{y}_0 - \sum_{k=1}^{d}\boldsymbol{v}_k\lambda_k^{-1}\boldsymbol{v}_k^\top\left(e^{-\lambda_k t} - 1\right)\boldsymbol{b}\right) \tag{114}$$

$$= \sum_{i=1}^{d}\boldsymbol{v}_i\left(\boldsymbol{v}_i^\top \boldsymbol{y}_0\right)e^{\lambda_i t} - \sum_{i=1}^{d}\boldsymbol{v}_i\left(\boldsymbol{v}_i^\top \boldsymbol{b}\right)\lambda_i^{-1}\left(1 - e^{\lambda_i t}\right)\,. \tag{115}$$

$\square$

Next, we prove Theorem B.2.

*Proof.* We assume WLOG that $\boldsymbol{x}_0 = 0$. Given the initial point $\boldsymbol{y}_0$ such that $\boldsymbol{y}_0$ such that $\rho < \boldsymbol{u}_i^\top \boldsymbol{y}_0 < \|\boldsymbol{x}_i\| - \rho$ and $\boldsymbol{u}_j^\top \boldsymbol{y}_0 < \rho$ for all $j \neq i$, the score is given by

$$s\left(\boldsymbol{y}\right) = \frac{1}{\sigma^2} \left( \frac{\|\boldsymbol{x}_i\|}{\|\boldsymbol{x}_i\| - 2\rho} \boldsymbol{u}_i \left( \boldsymbol{u}_i^\top \boldsymbol{y} - \rho \right) - \boldsymbol{y} \right) \tag{116}$$

$$= \frac{1}{\sigma^2} \left( \left( \frac{\|\boldsymbol{x}_i\|}{\|\boldsymbol{x}_i\| - 2\rho} \boldsymbol{u}_i \boldsymbol{u}_i^\top - \boldsymbol{I} \right) \boldsymbol{y} - \frac{\|\boldsymbol{x}_i\| \rho}{\|\boldsymbol{x}_i\| - 2\rho} \boldsymbol{u}_i \right). \tag{117}$$

According to Lemma D.4, the score flow in the partition $\rho < \boldsymbol{u}_i^\top \boldsymbol{y} < \|\boldsymbol{x}_i\| - \rho$ and $\boldsymbol{u}_j^\top \boldsymbol{y} < \rho$ for all $j \neq i$ is

$$\boldsymbol{y}_t = \sum_{k=1}^{d} \boldsymbol{v}_k \left( \boldsymbol{v}_k^\top \boldsymbol{y}_0 \right) e^{\lambda_k \frac{t}{\sigma^2}} - \sum_{k=1}^{d} \boldsymbol{v}_k \left( \boldsymbol{v}_k^\top \boldsymbol{b} \right) \lambda_k^{-1} \left( 1 - e^{\lambda_k \frac{t}{\sigma^2}} \right), \tag{118}$$

where the matrix $\boldsymbol{A} = \left( \frac{\|\boldsymbol{x}_i\|}{\|\boldsymbol{x}_i\| - 2\rho} \boldsymbol{u}_i \boldsymbol{u}_i^\top - \boldsymbol{I} \right)$. The eigenvalue decomposition of $\boldsymbol{A}$ is

$$\boldsymbol{A} = \boldsymbol{V} \boldsymbol{\Lambda} \boldsymbol{V}^\top \tag{119}$$

$$\boldsymbol{V} = \begin{pmatrix} \boldsymbol{u}_i & \boldsymbol{w}_1 & \cdots & \boldsymbol{w}_{d-1} \end{pmatrix} \tag{120}$$

$$\boldsymbol{\Lambda} = \text{diag}\left( \frac{2\rho}{\|\boldsymbol{x}_i\| - 2\rho}, -1, \cdots, -1 \right), \tag{121}$$

where $\boldsymbol{w}_j \in \boldsymbol{u}_i^\perp$. Since,

$$\left( \frac{\|\boldsymbol{x}_i\|}{\|\boldsymbol{x}_i\| - 2\rho} \boldsymbol{u}_i \boldsymbol{u}_i^\top - \boldsymbol{I} \right) \boldsymbol{u}_i = \left( \frac{\|\boldsymbol{x}_i\|}{\|\boldsymbol{x}_i\| - 2\rho} - 1 \right) \boldsymbol{u}_i \tag{122}$$

$$= \frac{2\rho}{\|\boldsymbol{x}_i\| - 2\rho} \boldsymbol{u}_i \tag{123}$$

$$\left( \frac{\|\boldsymbol{x}_i\|}{\|\boldsymbol{x}_i\| - 2\rho} \boldsymbol{u}_i \boldsymbol{u}_i^\top - \boldsymbol{I} \right) \boldsymbol{w}_j = -\boldsymbol{w}_j, \tag{124}$$

and $\boldsymbol{b} = -\frac{\|\boldsymbol{x}_i\| \rho}{\|\boldsymbol{x}_i\| - 2\rho} \boldsymbol{u}_i$. So, we get

$$\boldsymbol{y}_t = \boldsymbol{u}_i \left( \left( \boldsymbol{u}_i^\top \boldsymbol{y}_0 \right) e^{\frac{2\rho}{\|\boldsymbol{x}_i\| - 2\rho} \frac{t}{\sigma^2}} + \frac{\|\boldsymbol{x}_i\|}{2} \left( 1 - e^{\frac{2\rho}{\|\boldsymbol{x}_i\| - 2\rho} \frac{t}{\sigma^2}} \right) \right) + \sum_{k=2}^{d} \boldsymbol{v}_k \left( \boldsymbol{v}_k^\top \boldsymbol{y}_0 \right) e^{-\frac{t}{\sigma^2}}. \tag{125}$$

Note that we can analyze the score flow along each orthogonal direction separately. Next, we divide it into the following cases:

If $\boldsymbol{u}_i^\top \boldsymbol{y}_0 = \frac{\|\boldsymbol{x}_i\|}{2}$, then

$$\boldsymbol{y}_t = \boldsymbol{u}_i \frac{\|\boldsymbol{x}_i\|}{2} + \sum_{k=2}^{d} \boldsymbol{v}_k \left( \boldsymbol{v}_k^\top \boldsymbol{y}_0 \right) e^{-\frac{t}{\sigma^2}}. \tag{126}$$

Therefore, we converge to the point $\boldsymbol{y}_\infty = \boldsymbol{u}_i \frac{\|\boldsymbol{x}_i\|}{2}$.

If $\boldsymbol{u}_i^\top \boldsymbol{y}_0 > \frac{\|\boldsymbol{x}_i\|}{2}$, then we converge to $\boldsymbol{y}_\infty = \boldsymbol{u}_i \|\boldsymbol{x}_i\|$, and if $\boldsymbol{u}_i^\top \boldsymbol{y}_0 < \frac{\|\boldsymbol{x}_i\|}{2}$ then we converge to $\boldsymbol{y}_\infty = \boldsymbol{x}_1 = 0$ (since then the score function is $\frac{\|\boldsymbol{x}_i\| - \boldsymbol{y}}{\sigma^2}$ or $-\frac{\boldsymbol{y}}{\sigma^2}$).

We assume WLOG that $\boldsymbol{u}_i^\top \boldsymbol{y}_0 > \frac{\|\boldsymbol{x}_i\|}{2}$. We define $\Delta T_{\boldsymbol{u}_i}(\rho)$ time to reach the edge of the partition, i.e. $\|\boldsymbol{x}_i\| - \rho$ starting from the initialization point, and $\Delta T_{\boldsymbol{v}_k}(\rho, \epsilon)$ time to reach $\epsilon$ distance from zero (the data manifold) starting from the initialization point.

$$\Delta T_{\boldsymbol{u}_i}(\rho) = \sigma^2 \frac{\|\boldsymbol{x}_i\| - 2\rho}{2\rho} \log \left( \frac{\frac{\|\boldsymbol{x}_i\|}{2} - \rho}{\boldsymbol{u}_i^\top \boldsymbol{y}_0 - \frac{\|\boldsymbol{x}_i\|}{2}} \right) \tag{127}$$

$$\Delta T_{\boldsymbol{v}_k}(\rho, \epsilon) = \sigma^2 \log \left( \frac{\boldsymbol{v}_k^\top \boldsymbol{y}_0}{\epsilon} \right). \tag{128}$$

Since $\rho = \alpha\sigma$, we get that

$$\Delta T_{u_i}(\rho) = \rho \frac{\|x_i\| - 2\rho}{2\alpha^2} \log\left(\frac{\frac{\|x_i\|}{2} - \rho}{u_i^\top y_0 - \frac{\|x_i\|}{2}}\right) \tag{129}$$

$$\Delta T_{v_k}(\rho, \epsilon) = \left(\frac{\rho}{\alpha}\right)^2 \log\left(\frac{v_k^\top y_0}{\epsilon}\right). \tag{130}$$

Similarly to D.2, we get that $\exists \rho_0(\epsilon) > 0$ such that $\forall \rho < \rho_0(\epsilon,)$

$$T_0 = \max_k \Delta T_{v_k}(\epsilon) < T < \Delta T_{u_i}(\rho). \tag{131}$$

$\square$

### D.6. Proof of Theorem B.3

*Proof.* The estimated score function at the initialization is

$$\sigma_t^2 s(y, t) = \left(\left(1 + \frac{2}{\|x_i\|}\rho_t\right) u_i u_i^\top - I\right) y - \rho_t u_i. \tag{132}$$

Next, we project the estimated score along $u_i$ and the orthogonal direction, so we get

$$u_i u_i^\top \sigma_t^2 s(y, t) = \left(\left(1 + \frac{2}{\|x_i\|}\rho_t\right) u_i u_i^\top - u_i u_i^\top\right) y - \rho_t u_i \tag{133}$$

$$= u_i \rho_t \left(\frac{2}{\|x_i\|} u_i^\top y - 1\right) \tag{134}$$

$$(I - u_i u_i^\top) \sigma_t^2 s(y, t) = (I - u_i u_i^\top)\left(\left(1 + \frac{2}{\|x_i\|}\rho_t\right) u_i u_i^\top - I\right) y - \rho_t (I - u_i u_i^\top) u_i \tag{135}$$

$$= \left(\left(1 + \frac{2}{\|x_i\|}\rho_t\right) u_i u_i^\top - I\right) y - \left(\left(1 + \frac{2}{\|x_i\|}\rho_t\right) u_i u_i^\top - u_i u_i^\top\right) y \tag{136}$$

$$= \left(\left(1 + \frac{2}{\|x_i\|}\rho_t\right) u_i u_i^\top - I\right) y - \frac{2}{\|x_i\|}\rho_t u_i u_i^\top \tag{137}$$

$$= \left(u_i u_i^\top - I\right) y. \tag{138}$$

Therefore, the projected score onto $u_i$ is $\frac{\rho_t\left(\frac{2}{\|x_i\|} u_i^\top y - 1\right)}{\sigma_t^2}$, and the projected score function onto $w_j \in u_i^\perp$ is $-\frac{w_j^\top y}{\sigma_t^2}$, so we get the same estimated score as in Theorem 4.4 (we can analyze the score flow along each orthogonal direction separately). Therefore, along $w_j$ we get

$$w_j^\top y_t = w_j^\top y_T e^{(\log\sqrt{t} - \log\sqrt{T})} = w_j^\top y_T e^{\left(\log\sqrt{\frac{t}{T}}\right)} = (y_T)_i \sqrt{\frac{t}{T}}. \tag{139}$$

So, we obtain $w_j^\top y_0 = 0$. Along $u_i$ we get

$$u_i^\top y_t = \frac{\|x_i\|}{2} + \left(u_i^\top y_T - \frac{\|x_i\|}{2}\right) e^{\frac{2}{\|x_i\|}(-\alpha\sqrt{t} + \alpha\sqrt{T})}, \tag{140}$$

so we obtain $w_j^\top y_0 = \frac{\|x_i\|}{2} + \left(u_i^\top y_T - \frac{\|x_i\|}{2}\right) e^{\frac{2\alpha\sqrt{T}}{\|x_i\|}}$. Then, if

$$T \geq \left(\frac{\|x_i\|}{2\alpha}\right)^2 \log^2\left(\frac{\frac{\|x_i\|}{2}}{(y_T)_i - \frac{\|x_i\|}{2}}\right), \tag{141}$$

we converge to the closest point in the set $\{\boldsymbol{x}_0, \boldsymbol{x}_i\}$ to the initialization point $\boldsymbol{y}_T$ since the estimated score is equal to $-\frac{\boldsymbol{y}}{\sigma_t^2}$ or $\frac{\|\boldsymbol{x}_i\| - \boldsymbol{y}}{\sigma_t^2}$ and we converge to $0$ or $\|\boldsymbol{x}_i\|$ (as in Theorem 4.4), and if

$$T < \left(\frac{\|x_i\|}{2\alpha}\right)^2 \log^2\left(\frac{\frac{\|x_i\|}{2}}{(\boldsymbol{y}_T)_i - \frac{\|x_i\|}{2}}\right), \tag{142}$$

we converge to a point on the line connecting $\boldsymbol{x}_0$ and $\boldsymbol{x}_i$. $\qquad\square$

### D.7. Poof of Proposition C.1

*Proof.* We assume WLOG that $\boldsymbol{x}_0 = 0$. Note that since the convex hull of the training points is an equilateral triangle, then $\|x_i\| = \|x\|$. Given the initial point $\boldsymbol{y}_0$ such that $i \in \{1, 2\} - \frac{\|\boldsymbol{x}\|}{2} + \rho < \boldsymbol{u}_i^\top \boldsymbol{y} < \|\boldsymbol{x}\| - \rho$ and $\boldsymbol{u}_3^\top \boldsymbol{y} < -\frac{\|\boldsymbol{x}\|}{2} + \rho$, the score is given by

$$s(\boldsymbol{y}) = \frac{1}{\sigma^2}\left(\frac{\|\boldsymbol{x}\|}{\frac{3}{2}\|\boldsymbol{x}\| - 2\rho} \sum_{i=1}^{2}\left(\boldsymbol{u}_i\boldsymbol{u}_i^\top \boldsymbol{y} + \frac{1}{2}\boldsymbol{x}_i - \boldsymbol{u}_i\rho\right) - \boldsymbol{y}\right) \tag{143}$$

$$= \frac{1}{\sigma^2}\left(\frac{\|\boldsymbol{x}\|}{\frac{3}{2}\|\boldsymbol{x}\| - 2\rho}(\boldsymbol{u}_1\boldsymbol{u}_1^\top + \boldsymbol{u}_2\boldsymbol{u}_2^\top) - \boldsymbol{I}\right)\boldsymbol{y} \tag{144}$$

$$+ \frac{1}{\sigma^2}\left(\frac{\|\boldsymbol{x}\|}{\frac{3}{2}\|\boldsymbol{x}\| - 2\rho}\left(\frac{1}{2}\|\boldsymbol{x}\| - \rho\right)\boldsymbol{u}_1 + \frac{\|\boldsymbol{x}\|}{\frac{3}{2}\|\boldsymbol{x}\| - 2\rho}\left(\frac{1}{2}\|\boldsymbol{x}\| - \rho\right)\boldsymbol{u}_2\right). \tag{145}$$

According to Lemma D.4, the score flow in the partition $i \in \{1, 2\} - \frac{\|\boldsymbol{x}\|}{2} + \rho < \boldsymbol{u}_i^\top \boldsymbol{y} < \|\boldsymbol{x}\| - \rho$ and $\boldsymbol{u}_3^\top \boldsymbol{y} < -\frac{\|\boldsymbol{x}\|}{2} + \rho$ is

$$\boldsymbol{y}_t = \sum_{k=1}^{2} \boldsymbol{v}_k\left(\boldsymbol{v}_k^\top \boldsymbol{y}_0\right) e^{\lambda_k \frac{t}{\sigma^2}} - \sum_{k=1}^{2} \boldsymbol{v}_k\left(\boldsymbol{v}_k^\top \boldsymbol{b}\right) \lambda_k^{-1}\left(1 - e^{\lambda_k \frac{t}{\sigma^2}}\right), \tag{146}$$

where the matrix $\boldsymbol{A} = \left(\frac{\|\boldsymbol{x}\|}{\frac{3}{2}\|\boldsymbol{x}\| - 2\rho}(\boldsymbol{u}_1\boldsymbol{u}_1^\top + \boldsymbol{u}_2\boldsymbol{u}_2^\top) - \boldsymbol{I}\right)$. The eigenvalue decomposition of $\boldsymbol{A}$ is

$$\boldsymbol{A} = \boldsymbol{V}\boldsymbol{\Lambda}\boldsymbol{V}^\top \tag{147}$$

$$\boldsymbol{V} = \left(\frac{\boldsymbol{u}_1 - \boldsymbol{u}_2}{\sqrt{2\left(1 - \boldsymbol{u}_1^\top \boldsymbol{u}_2\right)}} \quad \frac{\boldsymbol{u}_1 + \boldsymbol{u}_2}{\sqrt{2\left(1 + \boldsymbol{u}_1^\top \boldsymbol{u}_2\right)}}\right) \tag{148}$$

$$\boldsymbol{\Lambda} = \text{diag}\left(\frac{\|\boldsymbol{x}\|}{\frac{3}{2}\|\boldsymbol{x}\| - 2\rho}\left(1 - \boldsymbol{u}_1^\top \boldsymbol{u}_2\right) - 1, \frac{\|\boldsymbol{x}\|}{\frac{3}{2}\|\boldsymbol{x}\| - 2\rho}\left(1 + \boldsymbol{u}_1^\top \boldsymbol{u}_2\right) - 1\right), \tag{149}$$

since,

$$\left(\frac{\|\mathbf{x}\|}{\frac{3}{2}\|\boldsymbol{x}\| - 2\rho}(\boldsymbol{u}_1\boldsymbol{u}_1^\top + \boldsymbol{u}_2\boldsymbol{u}_2^\top) - \boldsymbol{I}\right)(\boldsymbol{u}_1 - \boldsymbol{u}_2) = \frac{\|\boldsymbol{x}\|}{\frac{3}{2}\|\boldsymbol{x}\| - 2\rho}(\boldsymbol{u}_1 + \boldsymbol{u}_2\boldsymbol{u}_2^\top \boldsymbol{u}_1 - \boldsymbol{u}_1\boldsymbol{u}_1^\top \boldsymbol{u}_2 - \boldsymbol{u}_2) - (\boldsymbol{u}_1 - \boldsymbol{u}_2)$$
$$\tag{150}$$

$$= \left(\frac{\|\boldsymbol{x}\|}{\frac{3}{2}\|\boldsymbol{x}\| - 2\rho}\left(1 - \boldsymbol{u}_2^\top \boldsymbol{u}_1\right) - 1\right)(\boldsymbol{u}_1 - \boldsymbol{u}_2) \tag{151}$$

$$\left(\frac{\|\mathbf{x}\|}{\frac{3}{2}\|\boldsymbol{x}\| - 2\rho}(\boldsymbol{u}_1\boldsymbol{u}_1^\top + \boldsymbol{u}_2\boldsymbol{u}_2^\top) - \boldsymbol{I}\right)(\boldsymbol{u}_1 + \boldsymbol{u}_2) = \frac{\|\boldsymbol{x}\|}{\frac{3}{2}\|\boldsymbol{x}\| - 2\rho}(\boldsymbol{u}_1 + \boldsymbol{u}_2\boldsymbol{u}_2^\top \boldsymbol{u}_1 + \boldsymbol{u}_1\boldsymbol{u}_1^\top \boldsymbol{u}_2 + \boldsymbol{u}_2) - (\boldsymbol{u}_1 + \boldsymbol{u}_2)$$
$$\tag{152}$$

$$= \left(\frac{\|\boldsymbol{x}\|}{\frac{3}{2}\|\boldsymbol{x}\| - 2\rho}\left(1 + \boldsymbol{u}_2^\top \boldsymbol{u}_1\right) - 1\right)(\boldsymbol{u}_1 + \boldsymbol{u}_2), \tag{153}$$

and $\boldsymbol{b} = \frac{\|\boldsymbol{x}\|}{\frac{3}{2}\|\boldsymbol{x}\| - 2\rho}\left(\frac{1}{2}\|\boldsymbol{x}\| - \rho\right)\boldsymbol{u}_1 + \frac{\|\boldsymbol{x}\|}{\frac{3}{2}\|\boldsymbol{x}\| - 2\rho}\left(\frac{1}{2}\|\boldsymbol{x}\| - \rho\right)\boldsymbol{u}_2$. We assume WLOG that,

$$\boldsymbol{u}_1 = \begin{pmatrix} 0 \\ 1 \end{pmatrix}, \quad \boldsymbol{u}_2 = \begin{pmatrix} \frac{\sqrt{3}}{2} \\ -\frac{1}{2} \end{pmatrix}, \quad \boldsymbol{u}_3 = \begin{pmatrix} -\frac{\sqrt{3}}{2} \\ -\frac{1}{2} \end{pmatrix}, \tag{154}$$

and we get

$$v_1 = \frac{1}{\sqrt{3}} (u_1 - u_2) \tag{155}$$

$$v_2 = u_1 + u_2 = -u3 \tag{156}$$

$$\lambda_1 = \frac{\frac{3}{2} \|x\|}{\frac{3}{2} \|x\| - 2\rho} - 1 > 0 \tag{157}$$

$$\lambda_2 = -\left(1 - \frac{\frac{1}{2} \|x\|}{\frac{3}{2} \|x\| - 2\rho}\right) < 0 \,. \tag{158}$$

$$y_t = \frac{1}{\sqrt{3}} (u_1 - u_2) \left(\frac{1}{\sqrt{3}} (u_1 - u_2)^\top y_0\right) e^{\left(\frac{\frac{3}{2}\|x\|}{\frac{3}{2}\|x\|-2\rho} - 1\right)\frac{t}{\sigma^2}} \tag{159}$$

$$+ (u_1 + u_2) \left((u_1 + u_2)^\top y_0\right) e^{-\left(1 - \frac{\frac{1}{2}\|x\|}{\frac{3}{2}\|x\|-2\rho}\right)\frac{t}{\sigma^2}} \tag{160}$$

$$- (u_1 + u_2) \left(\frac{\|x\|}{\frac{3}{2}\|x\|-2\rho}\frac{1}{2}\|x\| - \rho\right) \left(\frac{\frac{1}{2}\|x\|}{\frac{3}{2}\|x\|-2\rho} - 1\right)^{-1} \left(1 - e^{-\left(1 - \frac{\frac{1}{2}\|x\|}{\frac{3}{2}\|x\|-2\rho}\right)\frac{t}{\sigma^2}}\right) \,. \tag{161}$$

Note that,

$$\left(\frac{\|x\|}{\frac{3}{2}\|x\|-2\rho}\left(\frac{1}{2}\|x\|-\rho\right)\right)\left(\frac{\frac{1}{2}\|x\|}{\frac{3}{2}\|x\|-2\rho}-1\right)^{-1} = \frac{\|x\|\left(\frac{1}{2}\|x\|-\rho\right)}{\frac{3}{2}\|x\|-2\rho}\frac{\frac{3}{2}\|x\|-2\rho}{-\|x\|+2\rho} \tag{162}$$

$$= \frac{\|x\|\left(\frac{1}{2}\|x\|-\rho\right)}{-\|x\|+2\rho} = -\frac{\|x\|}{2} \,. \tag{163}$$

Therefore,

$$y_t = \frac{1}{\sqrt{3}} (u_1 - u_2) \left(\frac{1}{\sqrt{3}} (u_1 - u_2)^\top y_0\right) e^{\left(\frac{\frac{3}{2}\|x\|}{\frac{3}{2}\|x\|-2\rho} - 1\right)\frac{t}{\sigma^2}} \tag{164}$$

$$+ (u_1 + u_2) \left((u_1 + u_2)^\top y_0\right) e^{-\left(1 - \frac{\frac{1}{2}\|x\|}{\frac{3}{2}\|x\|-2\rho}\right)\frac{t}{\sigma^2}} \tag{165}$$

$$- (u_1 + u_2) \left(-\frac{\|x\|}{2}\right) \left(1 - e^{-\left(1 - \frac{\frac{1}{2}\|x\|}{\frac{3}{2}\|x\|-2\rho}\right)\frac{t}{\sigma^2}}\right) \tag{166}$$

$$= \frac{1}{\sqrt{3}} (u_1 - u_2) \left(\frac{1}{\sqrt{3}} (u_1 - u_2)^\top y_0\right) e^{\left(\frac{\frac{3}{2}\|x\|}{\frac{3}{2}\|x\|-2\rho} - 1\right)\frac{t}{\sigma^2}} \tag{167}$$

$$+ (u_1 + u_2) \left(\left((u_1 + u_2)^\top y_0 - \frac{\|x\|}{2}\right) e^{-\left(1 - \frac{\frac{1}{2}\|x\|}{\frac{3}{2}\|x\|-2\rho}\right)\frac{t}{\sigma^2}} + \frac{\|x\|}{2}\right) \,. \tag{168}$$

Note that we can analyze the score flow along each orthogonal direction separately. Next, we divide it into the following cases:

If $\frac{1}{\sqrt{3}} (u_1 - u_2)^\top y_0 = 0$, then

$$y_t = (u_1 + u_2) \left(\left((u_1 + u_2)^\top y_0 - \frac{\|x\|}{2}\right) e^{-\left(1 - \frac{\frac{1}{2}\|x\|}{\frac{3}{2}\|x\|-2\rho}\right)\frac{t}{\sigma^2}} + \frac{\|x\|}{2}\right) \,, \tag{169}$$

and we converge to the point $y_\infty = (u_1 + u_2) \frac{\|x\|}{2}$.

If $\frac{1}{\sqrt{3}} (u_1 - u_2)^\top y_0 > 0$, then we converge to $y_\infty = x_1$, and if $\frac{1}{\sqrt{3}} (u_1 - u_2)^\top y_0 < 0$, then we converge to $y_\infty = x_2$.

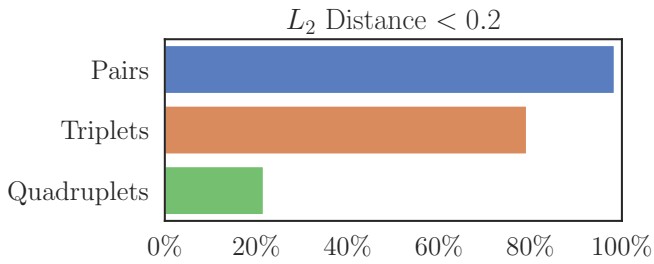

Figure 6: **Existence of stable virtual training points.** We run fixed-point iterations on a single denoiser, starting from all possible pair-wise, triplet-wise, and quadruplet-wise combinations of training samples. The plot shows the percentage of points that converged within an $L_2$ distance of 0.2 to the original, virtual, input point.

We assume WLOG that $\frac{1}{\sqrt{3}} (\boldsymbol{u}_1 - \boldsymbol{u}_2)^\top \boldsymbol{y}_0 > 0$. We define $\Delta T_d (\rho, \epsilon)$ as the time to reach $\epsilon$ distance from the data manifold (the line connecting the training points $\boldsymbol{x}_1$ and $\boldsymbol{x}_2$) starting from initialization point $\boldsymbol{y}_0$, and $\Delta T_e (\rho)$ the time to reach the edge of the partition starting from initialization point $\boldsymbol{y}_0$. We assume WLOG that $(\boldsymbol{u}_1 + \boldsymbol{u}_2)^\top \boldsymbol{y}_0 > \frac{\|\boldsymbol{x}\|}{2}$ and $(\boldsymbol{u}_1 + \boldsymbol{u}_2)^\top \boldsymbol{y}_0 - \frac{\|\boldsymbol{x}\|}{2} > \epsilon$

$$\Delta T_d (\rho, \epsilon) = \frac{\sigma^2}{\frac{\frac{1}{2}\|\boldsymbol{x}\|}{\frac{3}{2}\|\boldsymbol{x}\|-2\rho} - 1} \log \left( \frac{\epsilon}{(\boldsymbol{u}_1 + \boldsymbol{u}_2)^\top \boldsymbol{y}_0 - \frac{\|\boldsymbol{x}\|}{2}} \right) \tag{170}$$

$$\Delta T_e (\rho) = \frac{\sigma^2}{\frac{\frac{3}{2}\|\boldsymbol{x}\|}{\frac{3}{2}\|\boldsymbol{x}\|-2\rho} - 1} \log \left( \frac{\frac{1}{2}\|\boldsymbol{x}\| - \rho}{\frac{1}{\sqrt{3}}(\boldsymbol{u}_1 - \boldsymbol{u}_2)^\top \boldsymbol{y}_0} \right). \tag{171}$$

Since $\rho = \alpha\sigma$, we get that

$$\Delta T_d (\rho, \epsilon) = \frac{\rho^2}{\alpha^2 \left( \frac{\frac{1}{2}\|\boldsymbol{x}\|}{\frac{3}{2}\|\boldsymbol{x}\|-2\rho} - 1 \right)} \log \left( \frac{\epsilon}{(\boldsymbol{u}_1 + \boldsymbol{u}_2)^\top \boldsymbol{y}_0 - \frac{\|\boldsymbol{x}\|}{2}} \right) \tag{172}$$

$$\Delta T_e (\rho) = \frac{\rho^2}{\alpha^2 \left( \frac{\frac{3}{2}\|\boldsymbol{x}\|}{\frac{3}{2}\|\boldsymbol{x}\|-2\rho} - 1 \right)} \log \left( \frac{\frac{1}{2}\|\boldsymbol{x}\| - \rho}{\frac{1}{\sqrt{3}}(\boldsymbol{u}_1 - \boldsymbol{u}_2)^\top \boldsymbol{y}_0} \right). \tag{173}$$

Similar to D.2 we get that $\exists \rho_0 (\epsilon) > 0$ such that $\forall \rho < \rho_0 (\epsilon, )$

$$T_0 = \Delta T_d (\rho, \epsilon) < T < T_1 = \Delta T_e (\rho) . \tag{174}$$

$\square$

## E. Exploration of Different Thresholds

We next repeat the statistical analysis done in Section 5 for different thresholds. Figure 6 demonstrates the existence of virtual points, in an analogous way to Figure 1, for the $L_2$ metric. Figures 7 and 8 offer additional insights to the right side of Figure 2a. Specifically, in Figure 7 we compare the results of the convergence types frequency of randomly sampled points with score flow when using different thresholds of the $L_\infty$ distance. In Figure 8 we instead use the $L_2$ metric. Similarly, Figures 9 and 10 depict additional comparisons to the right side of Figure 2b, for both the $L_\infty$ and $L_2$ distance metrics.

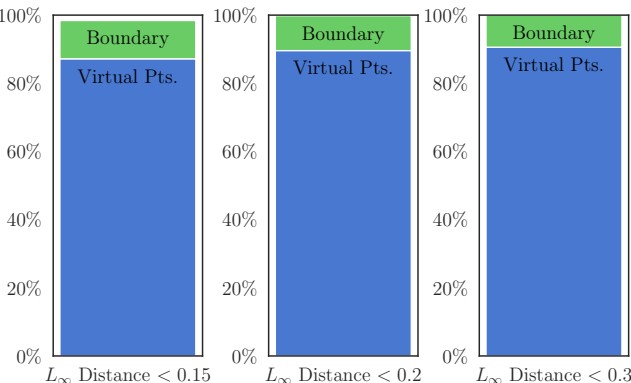

Figure 7: **Convergence types frequency of randomly sampled points for score flow based on $L_\infty$ proximity.** We run the discrete ODE formulation of equation 20 for 500 randomly sampled points from $\mathbb{R}^{30}$ for sampling using the score flow. We plot the percentage of points that converged to either a virtual point, a training point, or to the boundaries of the hyperbox, out of all points, based on their $L_\infty$ proximity for different thresholds.

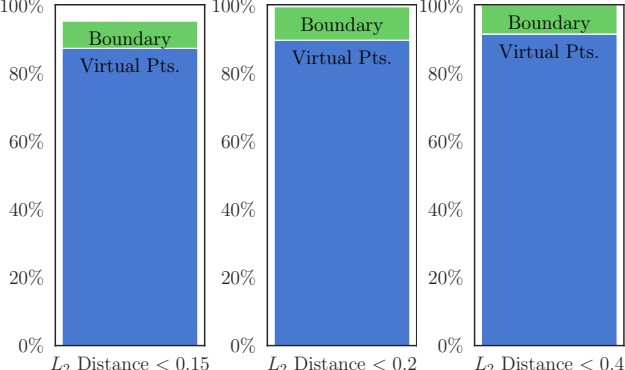

Figure 8: **Convergence types frequency of randomly sampled points for score flow based on $L_2$ proximity.** We run the discrete ODE formulation of equation 20 for 500 randomly sampled points from $\mathbb{R}^{30}$ for sampling using the score flow. We plot the percentage of points that converged to either a virtual point, a training point, or to the boundaries of the hyperbox, out of all points, based on their $L_2$ proximity for different thresholds.

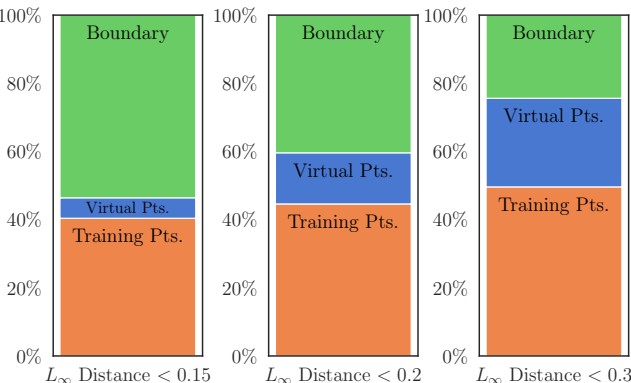

Figure 9: **Convergence types frequency of randomly sampled points for probability flow based on $L_\infty$ proximity.** We run the discrete ODE formulation of equation 20 for 500 randomly sampled points from $\mathbb{R}^{30}$ for probability flow. We plot the percentage of points that converged to either a virtual point, a training point, or to the boundaries of the hyperbox, out of all points, based on their $L_\infty$ proximity for different thresholds.

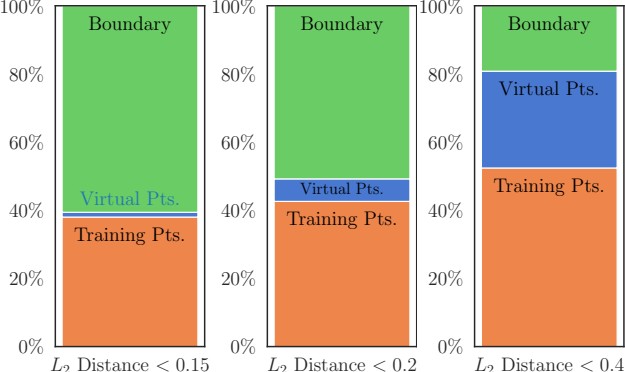

Figure 10: **Convergence types frequency of randomly sampled points for probability flow based on $L_2$ proximity.** We run the discrete ODE formulation of equation 20 for 500 randomly sampled points from $\mathbb{R}^{30}$ for probability flow. We plot the percentage of points that converged to either a virtual point, a training point, or to the boundaries of the hyperbox, out of all points, based on their $L_2$ proximity for different thresholds.

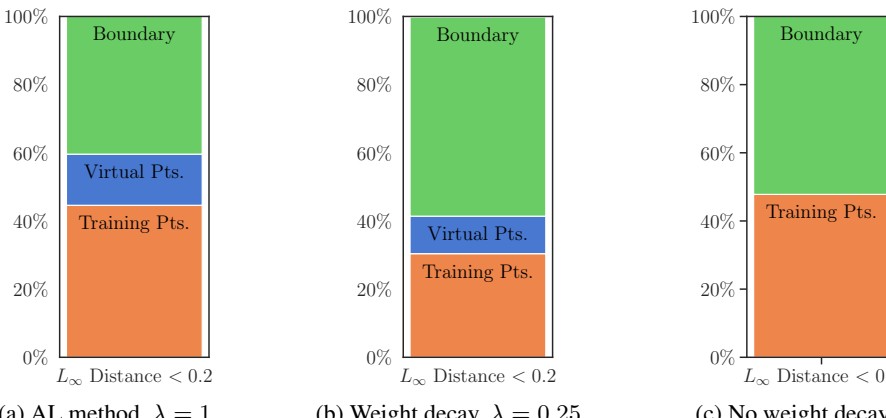

(a) AL method, $\lambda = 1$. (b) Weight decay, $\lambda = 0.25$. (c) No weight decay.

Figure 11: **Convergence types frequency of randomly sampled points in diffusion sampling for training with AL method, weight decay, and without weight decay.** We run the discrete ODE formulation of equation 20 for 500 randomly sampled points from $\mathbb{R}^{30}$ for diffusion sampling, using different training configurations. We plot the percentage of points that converged to either a virtual point, a training point, or to the boundaries of the hyperbox, out of all points. The minimum norm constraint is necessary for inducing the bias towards virtual training points and the boundaries of the hyperbox. Additionally, standard training protocol using weight decay regularization simulates well the minimum norm denoiser, which is achieved by the use of the AL method.

## F. The Minimum Norm Assumption

Theorems C.1, 4.4, B.1, B.2 and B.3 all hold in the case of a minimum norm denoiser, in which the denoiser achieves exact interpolation over the noisy training samples. To enforce a consistent denoiser, we used a non-standard training protocol in Section 5. Specifically, we optimize an equality-constrained optimization problem using the Augmented Lagrangian method. Here we verify the robustness of our results and the necessity of the minimum norm assumption by repeating the experiment from Section 5 when using standard training, with and without the use of weight decay. Specifically, all the hyper parameters and Adam optimizer are kept the same, and only the loss function changes to directly optimize equation 3. Training with weight decay should result in a denoiser that is similar to the min-norm solution. Figure 11 shows the percentage of points that converged within an $L_\infty$ distance of 0.2 to either virtual points, training points, or a boundary of the hyperbox, for the different training configurations. The use of weight decay in a standard training protocol induces a similar bias to that achieved by the use of Augmented Lagrangian method.

## G. Extension to Orthogonal Data With $N > d + 1$

As in the case of strictly orthogonal data it is impossible to have $N > d$, we consider here a simliar setup where the training set is augmented with samples randomly drawn from the boundary of the hyperbox. Specifically, for each augmented sample we sample a random vector with i.i.d. elements from the uniform distribution $\mathrm{Unif}[0.3, 0.7]$. This choice avoids the degenerate case where no denoisers are active in the low-noise regime. We then project the vector on a random face of the hyperbox to ensure that the new random data points lie on the hyperbox boundary. We train the denoisers following the same experimental setup as in Figure 2, using $M = 500$ noisy samples and the AL method for optimization. We report the the numerical values for the convergence of points for the $L_\infty$ metric with a 0.2 threshold in Table 1. As can be seen from these results, in cases where $N > d + 1$ the probability flow almost exclusively converges to the hyperbox boundary. Specifically, either to the boundary, training points (either old orthogonal points or the new points on the boundary), or to other vertices of the hyperbox (the original virtual points), which aligns with our theory.

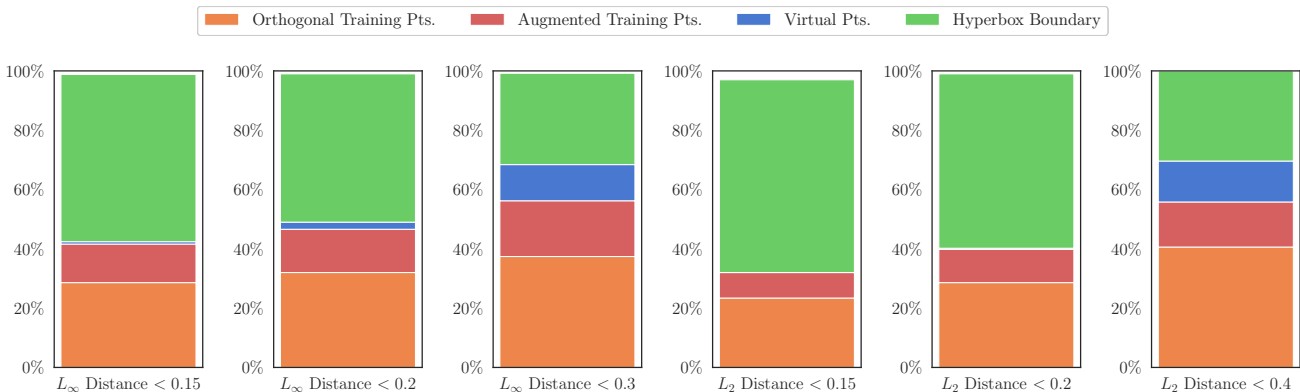

Figure 12: **Convergence types frequency of randomly sampled points in diffusion sampling for training with AL method, with an augmented training set such that $N > d + 1$, where $N = 40$.** We run the discrete ODE formulation of equation 20 for 500 randomly sampled points from $\mathbb{R}^{30}$ for probability flow. We plot the percentage of points that converged to either a virtual point, a training point, a new augmented training point from the boundary of the hyperbox, or to the boundaries of the hyperbox, out of all points, based on their $L_\infty$ or $L_2$ proximity for different thresholds.

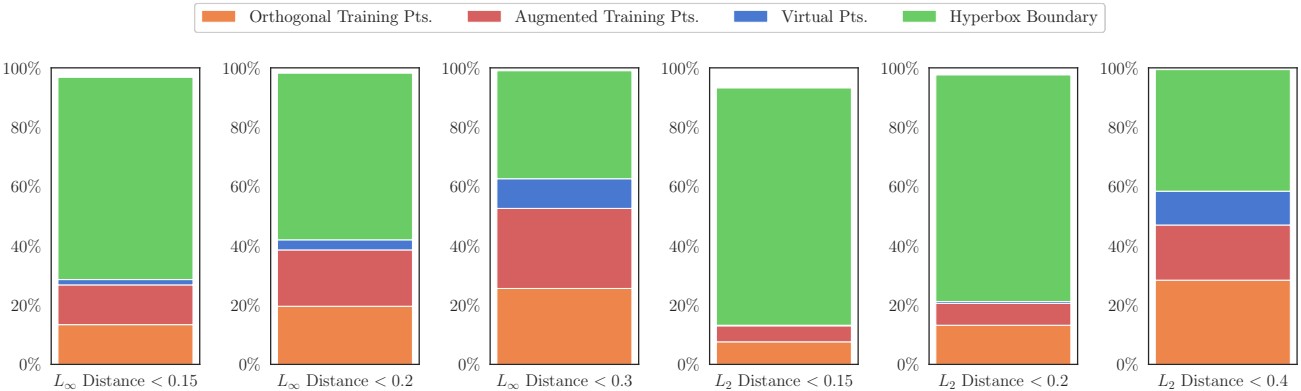

Figure 13: **Convergence types frequency of randomly sampled points in diffusion sampling for training with AL method, with an augmented training set such that $N > d$, where $N = 50$.** We run the discrete ODE formulation of equation 20 for 500 randomly sampled points from $\mathbb{R}^{30}$ for probability flow. We plot the percentage of points that converged to either a virtual point, a training point, a new augmented training point from the boundary of the hyperbox, or to the boundaries of the hyperbox, out of all points, based on their $L_\infty$ or $L_2$ proximity for different thresholds.

Table 1: Convergence types frequency of randomly sampled points in diffusion sampling for training with AL method, with an augmented training set such that $N > d + 1$.

|  | $N = 40$ | $N = 50$ |
|---|---|---|
| **Hyperbox Boundary** | 99% | 98.21% |
| **Original Virtual Points** | 2.4% | 3.4% |
| **Orthogonal Training Datapoints** | 32% | 19.6% |
| **Augmented Random Data Points** | 14.6% | 19% |

Further metrics and thresholds for the two cases can be seen in Figures 12 and 13.

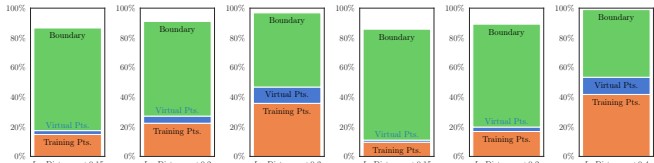

Figure 14: **Convergence types frequency of randomly sampled points in diffusion sampling for training with AL and dropout.** We run the discrete ODE formulation of equation 20 for 500 randomly sampled points from $\mathbb{R}^{30}$ for probability flow. We plot the percentage of points that converged to either a virtual point, a training point, a new augmented training point from the boundary of the hyperbox, or to the boundaries of the hyperbox, out of all points, based on their $L_\infty$ or $L_2$ proximity for different thresholds.

## H. Including Dropout During Training

We conduct additional experiments to assess the effect of incorporating dropout during training. As shown in Figure 14, this results in more samples being generated outside the boundary of the hyperbox. Notably, adding dropout increases the MSE loss during denoiser training, especially in the "low-noise regime", which contributes to this behavior.

## I. Connection to Hopfield Models

We compute the detection metrics suggested by Pham et al. (2024) for memorization, spurious, and generalization detection, for the same setup as in Figure 2. Specifically, the training set contains $N = 31$ orthogonal points, and we use the same 500 sampled points as the evaluation set. Therefore, in the notation of Pham et al. (2024), $|S| = 31$ and $S^{\text{eval}}| = 500$. Following the guidelines set by Pham et al. (2024), we ensure the additional required set $S'$ is much larger than the training set by constructing $S'$ such that $|S'| = 100 \times |S|$. Additionally, both $S'$ and $S^{\text{eval}}$ were sampled using probability flow.

Table 2 compares the classification of evaluation points into memorization, spurious, and generalization categories as described by Pham et al. (2024), with our own categorization into training points, virtual points, and hyperbox boundary points. We set $\delta_m = L$, corresponding to the thresholds used in our experiments, and test both $L_\infty$ and $L_2$. Additionally, we use $\delta_s = 0.15$. The results suggest that many virtual points are classified as spurious under the criteria set by Pham et al. (2024). This aligns with our analysis, as virtual points are stable, stationary points of the score flow; therefore, given a large evaluation set, points will cluster near the virtual points, which matches the spurious points definition. However, due to the exponential number of virtual points, we cannot expect a cluster to form around each of them. As a result, some virtual points are also classified as "generalization" under these metrics.

Table 2: Comparison of the classification of evaluation points between our notations and the notations of Pham et al. (2024).

| Classification under Pham et al. (2024) | Our Classification | $L_\infty = 0.2$ | $L_\infty = 0.15$ | $L_\infty = 0.3$ | $L_2 = 0.2$ | $L_2 = 0.4$ |
|---|---|---|---|---|---|---|
| **Memorized** | Training Pts. | 100% | 99.49% | 100% | 100% | 100% |
| | Boundary Pts. | 0% | 0.51% | 0% | 0% | 0% |
| **Spurious** | Virtual Pts. | 52.94% | 21.57% | 78.22% | 34.29% | 100% |
| | Training Pts. | 0.98% | 0% | 0% | 0% | 0% |
| | Boundary Pts. | 46.08% | 78.43% | 21.78% | 65.71% | 0% |
| **Generalization** | Virtual Pts. | 11.35% | 3.98% | 31.48% | 3.98% | 40.22% |
| | Training Pts. | 4.86% | 2.99% | 6.79% | 3.98% | 6.15% |
| | Boundary Pts. | 83.78% | 93.03% | 64.73% | 92.04% | 53.63% |

## J. Additional Simulations

Figure 4 shows the normalized score flow for the case of an obtuse 2-simplex. Figure 5 shows the normalized score flow for the case of an equilateral triangle. The normalization was done for visualization purposes only, since the norm of the score

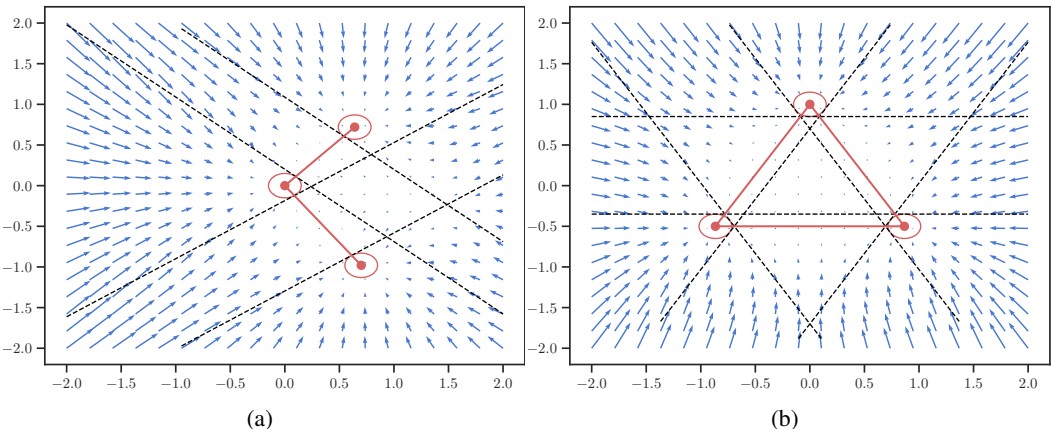

Figure 15: **The score function of obtuse and acute simplex**. The red dots are the training points $x_1, x_2, x_3$. The black lines are the ReLU boundaries. In figure (a) we plot the score function of obtuse simplex (Proposition D.1). In figure (b) we plot acute simplex (Proposition D.2)

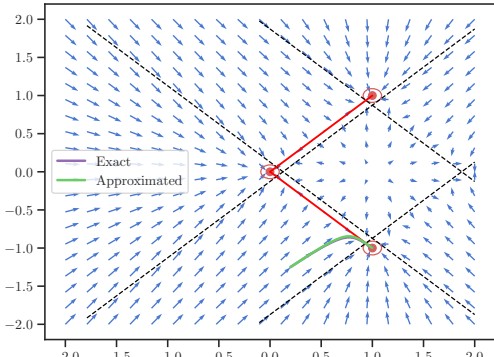

Figure 16: **The score function of orthogonal dataset**. The purple line is the trajectory of the score flow of the exact score function, and the green line is the trajectory of the score flow of the approximated score function (equation 18) in the case where $\sigma = 0.03$, $\rho = 0.09$. Both trajectories are very similar.

decreases as it approaches the ReLU boundaries. In Figure 15 we illustrate the unnormalized score flow. Figure 16 shows the trajectory of score flow of the exact score function, and the green line is trajectory of the score flow of the approximated score function as can be seen the trajectories are practically identical.

