# OpenReview forum: "When Diffusion Models Memorize: Inductive Biases in Probability Flow of Minimum-Norm Shallow Neural Nets"
_ICML.cc/2025/Conference — ICML 2025 poster_

### Official Review · Reviewer_vy6L · 2025-03-07

**Overall Recommendation:** 3

**Summary:**

The paper rigorously proves that, when the training points are orthogonal and we use the MMSE denoiser, score flow converges to the vertices of a hyperbox, wherein the vertices of the hyperbox are the partial sums of training points. Similarly, they prove that probability flow converges to a vertex on this hyperbox or a point on the boundary of the hyperbox. Both proofs analyze the stable points of the flow under a "low-noise regime" showing that, for score flow, the only stable points are the vertices of the hyperbox, whereas for probability flow these stable point include the boundary. They proceed to show that these results hold using a shallow ReLU neural network with a minimal l2 norm with some error wherein some partial sums have become unstable due to the error between the MMSE and neural network denoiser. Finally, they show experimentally that as the number of training points increases, the diffusion process converges to a higher proportion of non-training points thus the generalization is increasing.

**Claims And Evidence:**

All claims are stated clearly and have been rigorously proved. Any assumptions made for the theory are well discussed. The theory is then backed by sufficient experiments which extend to a more practical setting.

**Essential References Not Discussed:**

None that I am aware of.

**Experimental Designs Or Analyses:**

The main experiments are with carefully constructed data and models in line with the theory proposed.  There are supplementary experiments which relax the assumptions and demonstrate that the theory still hold with some error.

**Methods And Evaluation Criteria:**

The stable point analysis used to prove the claims gives a good intuition for convergence and the approximations used to simplify the analysis are justified.

**Other Comments Or Suggestions:**

- For the diagrams in Figure 3, if Virtual pts. were above Training pts., I think it would be easier to see that the generalization increases and that a larger percentage of samples converge to the virtual and boundary points.

**Other Strengths And Weaknesses:**

Strengths:
- The paper provides a rigorous analysis of the stable points of probability flow and the score flow, this contributes greatly to our understanding of where a diffusion process consisting of a sequence of denoisers converges to.
- The results focus on the impact that training data has on the final convergence points of the diffusion process; effectively explaining what forms the new data produced by the diffusion process can take on.
- The paper presents an interesting view that generalization in diffusion models is the result of having a larger proportion of stable points which are not the training data.
- The stable point analysis is nice and relatively easy to follow. The diagrams help demonstrate what is happening.
- Experiments align closely with the theory considered and help consolidate the conclusions.

Weaknesses:
- Analysis is restricted to the "low-noise regime", but this is not mentioned in the limitations section. Specifically, for every experiment the "low-noise regime" was supposedly enforced, but it is unclear how. Line 247 mentions 1/3 of the denoisers used are trained with this low-noise regime, but does not allude to how this might differ in practice. Are we guaranteed to have some denoisers in the "low-noise regime"?
- Experiments stick to synthetic data which is orthogonal. Since diffusion models are often used on image data, some mention of how this applies or can be understood on image data would be useful.
- The paper lacks discussion about the consequences of these results in a practical setting. The results may not precisely hold in such a setting, but what influence do they have? Do they provide intuition about the result of a diffusion process in general?
- The theoretical results rely heavily on Zeno et al. (NeurIPS 2023). This is unfortunate, as extending this work to alternative arrangements of data than those considered would require also extending the work of Zeno et al.

**Questions For Authors:**

1.  Do you have an intuition as to why we never converge inside the hyperbox? What would the data look like there? Is it not meaningful with respect to the training points?
2.  What intuition can your results tell us about diffusion models used in practice? For example, what does the hyperbox look like for image data, can the results provide intuition about the images diffusion models converges to?

**Relation To Broader Scientific Literature:**

- They move beyond Zhang & Pilanci's (arXiv 2024) work on univariate data by considering high-dimensional data belonging to a simplex and showing convergence using probability flow and score flow instead of Langevin Dynamics; thus showing different convergence guarantees depending on the inference procedure used.
- Carlini et al. (USENIX Security 2023) empirically showed that training data could be extracted by locating stable points. This work significantly advances this by providing theoretical guarantees that training points are stable points for certain arrangements of data (i.e., orthogonal data).
- There have been recent studies into the relationship between memorization and generalization. This work suggests that generalization improves with the number of training points, but memorization persists (training points are still stable but we are less likely to converge to them). This is a fascinating contribution as it is often thought that memorisation implies a lack of generalization.
- The theoretical results rely heavily on Zeno et al. (NeurIPS 2023) which provides an explicit form of the minimum MSE estimator for particular arrangements of data points. This is unfortunate, as extending this work to alternative arrangements than those considered would require also extending the work of Zeno et al. This point is also noted below in weaknesses.

**Theoretical Claims:**

I did not check the proofs in the appendix.

---

> ### Author Rebuttal · Authors · 2025-04-01
>
> We sincerely appreciate the reviewer’s positive feedback and are glad that they find this a fascinating contribution. The interplay between memorization and generalization is indeed intriguing.
> # Low-Noise Regime Details
> Theoretically, the boundary of the low-noise regime is determined by the diffusion time scheduler, the number of timesteps, $T$, the training data, and the noise realizations. In the experiments, we approximate this boundary by considering a ball radius equal to 4 standard deviations. Specifically, we first calculate the variance schedule of the $T=100$ diffusion timesteps. Then, we check the minimal distance between training points. We consider the low-noise regime boundary to be an eighth of this distance. Once we have this boundary, we extend the variance schedule to include at least $50$ variance values below the boundary.
> Since we are discretizing an ODE, as long as we are using small enough timesteps, we would get the same results. We will clarify this in the revised manuscript.
> In practical models, we are guaranteed to have some denoisers in the low-noise regime, since the final sampled point is obtained when $t \approx 0$, where the noise level is inherently small.
>
> Additionally, the low-noise regime is practically important for diffusion sampling, as the "useful" part of the diffusion dynamics occurs primarily when the noise level drops below a certain critical threshold [R1].
> # Implications for Practical Image Diffusion Models
> In practice, natural images lie on a low-dimensional linear subspace due to their approximate low-rank structure (see Zeno et al. (2023), Appendix D3). As a result, the denoiser first contracts the data towards this subspace (see Theorem 2 in Zeno et al. (2023)). If the subspace has sufficiently high dimension, the data points become approximately orthogonal (following the $\exp(D) \gg N > D \gg 1$ argument in our response to Reviewer SMjF under “Orthogonal Dataset Choice”). Consequently, the geometric structure in high-dimensional image spaces might resemble the hyperbox studied in our work, though restricted to a subspace, and with slightly acute angles rather than strictly orthogonal ones.
>
> # Reliance on Zeno et al. (NeurIPS 2023)
> We acknowledge the reliance on Zeno et al. (NeurIPS 2023), but we believe that advancing the understanding of diffusion models remains valuable, even within these constraints. We believe that exploring alternative data arrangements is an interesting and challenging direction for future work.
> # Fig. 3 Points Order
> Thanks for this valuable suggestion, we will swap the order in the revised paper.
> # Convergence Inside the Hyper-Box
> We do not converge inside the hyperbox because, as shown in Fig. 8 (Appendix F), the score function trajectory, both inside and outside the hyperbox, always points toward its boundary. As a result, the probability flow ODE leads to convergence at the hyperbox boundary rather than within its interior.
>
> [R1] Gabriel Raya & Luca Ambrogioni. Spontaneous symmetry breaking in generative diffusion models. NeurIPS, 2023.

---

### Official Review · Reviewer_urF4 · 2025-03-13

**Overall Recommendation:** 4

**Summary:**

Commonly, diffusion models use the probability flow ODE to generate high-quality images. Obviously, the score (or reverse) SDE can be used to generate images as well. But the process for the score SDE is understandable, since it is derived from the forward SDE. However, it is unclear on the distinction between the probability flow and the score SDE, regarding their ability to generate high quality images and duplicate training points. To tackle this distinction and to further elucidate on the mechanism of generating from probability flow ODE, this work studies the probability flow of shallow ReLU neural network denoisers trained with minimal $\ell_2$ norm on datasets, where the actual minimizers can be solved. Interestingly, the work finds that probability flow reaches more general manifold points, while score flow (SF) more often converges to training points or virtual points (sum of training points).

**Claims And Evidence:**

## Claim

The claims made by this paper are theoretical and backed by the simulation results on simple-solvable datasets in conjunction with proofs provided by the authors. These claims are:

(1) On clean and orthogonal data, both score and probability flow samplings follow a similar trajectory for a given initialization point.

(2) However, sampling with probability flow can reach more general manifold points due to the early stopping in the scheduler. In other words, sampling with probability flow can reach both training points and virtual points (which are sums of the training points). In the hyper-box data, such virtual points can exist on the boundary of the hyper-box. Although score flow ODE can converge to training points, it is quite rare for that to happen. In contrast, it is more likely to converge on virtual points. Meanwhile, probability flow ODE tend to converge to virtual points at the boundary.


## Evidence

(1) To establish the first claim, strongly following the work of [Zeno et al. (2023)](https://openreview.net/forum?id=gdzxWGGxWE), the authors first define the problem setting and the representation cost $R(\mathbf{h})$ where $\mathbf{h}$ is the **predicted training point** given a perturbation $\mathbf{y}_t$ of a training point $\mathbf{x}_i \in \{ \mathbf{x}_n \}^{N}_{n = 1}$ in which it tries to predict.

This representation cost $R$ focuses on the minimization of the $\ell_2$ regularization term $C(\theta)$ from the following MSE loss:

$\mathcal{L} (\theta) = \frac{1}{M N} \sum^M_{m = 1} \sum^N_{n = 1} \lVert \mathbf{h} (\mathbf{y}_{n, m}) - \mathbf{x}_n  \rVert^2  + \lambda C(\theta)$

where $\mathbf{y}_{n,m} = \mathbf{x}_n + \epsilon_{n, m}$ in which there are $m = 1, \dots, M$ perturbations and $n = 1, \dots, N$ training samples.

The minimizer of $R$ is: $\quad \quad \mathbf{h}^* \in \underset{\mathbf{h}}{argmin} \ R(\mathbf{h}) \quad $ s.t. $\quad \mathbf{h}(\mathbf{y}_{n, m}) = \mathbf{x}_n \quad \forall n, m$

but for the multivariate (more general) case, the constraint $\mathbf{h}(\mathbf{y}_{n, m}) = \mathbf{x}_n$ is modified into $\mathbf{h}(B(\mathbf{x}_n, p)) = \{ \mathbf{x}_n \}$. $B(\mathbf{x}_n, p)$ denotes a ball centered at the training point $\mathbf{x}_n$ with radius $p$. Thus, the minimizer setup, which the analysis focuses on, is

$\mathbf{h}_p^* \in \underset{\mathbf{h}}{argmin} \ R(\mathbf{h}) \quad $ s.t. $\quad \mathbf{h}(B(\mathbf{x}_n, p)) = \{ \mathbf{x}_n \} \quad \forall n$

Detailed in section (4), using the hyper-box as the data, the authors demonstrated the score flow converges to the vertex (or training point) of the hyper-box closest to its initialization point (see Theorem 4.3 and Appendix D.2). Based on Eq. (18), it is demonstrated that both of the flows share similar trajectory, in which they can arrive at similar stationary points.


(2) In the case of probability flow on the hyper-box, there is a $p_t$​ scheduler which inhibits its guarantee to land on the closest vertex. Thus, it can instead more often land on a boundary point to the vertex. **This directly supports claim (2) that probability flow can reach more general manifold points due to early stopping in the scheduler**. The authors formalize this through theoretical analysis showing that while score flow ODE tends to converge to virtual points (or sums of training points), probability flow ODE has a higher likelihood of converging to points on the boundary of the data manifold.

The empirical evidence in Figure (2) clearly illustrates this distinction, showing that probability flow sampling more frequently lands on boundary points, whereas score flow predominantly lands on virtual points. This systematic difference in convergence behavior confirms the theoretical prediction that probability flow can reach both training points and a broader set of manifold points (including boundary points), while score flow is biased toward virtual points.

These aspects are also established in two more data settings, obtuse-angle dataset and equilateral triangle dataset.

(3) The convergence to training points for both flows is further exacerbated as there are more training points. Partly, this is due to the creation of more virtual points. This fact is demonstrated in Figure (3) which details the fraction of training points, boundary points, and virtual points as the training data size $N$ increases. Although the figure only shows up to $N = 30$, as $N \rightarrow \infty$ it is likely that there will be more virtual points than training data generated from the models.

## Strength

(1) The methodology of using solvable shallow ReLU networks allows for precise mathematical characterization of the sampling behaviors, leading to provable results rather than just empirical observations. This approach enables the authors to establish clear connections between score and probability flows and their trajectory in generation.

(2) The findings regarding virtual points and boundary points have significant implications for diffusion model's dynamics. Specifically, by revealing how probability flow can reach more general manifold points while score flow more often converges to training or virtual points, the paper provides some insights on memorization and generalization in diffusion models.

(3) Overall, the paper is presented in an accessible and engaging manner, which makes it approachable for both newcomers to the field and experts. Frankly, I did enjoy reading the paper.

## Weakness

(1) The paper's analysis is limited to simplified data distributions and a shallow ReLU network (with a single hidden layer), which may not generalize well to complex, high-dimensional data distributions used in practical applications. This restriction to tractable theoretical settings leaves questions about whether the conclusions apply to modern deep diffusion models which operate at a much higher dimension and learn more complex distributions.

(2) Furthermore, the analysis of this work relies heavily on the work done by [Zeno et al. (2023)](https://openreview.net/forum?id=gdzxWGGxWE), which assumes a full $d-$ dimensional ball centered around each training data. In real-world scenarios, the number of noisy samples is typically smaller than the input dimension $d$.

(3) Moreover, the paper's analysis did not include manifold with curvatures. Many real-world datasets, especially in high-dimensional spaces, exhibit complex geometric properties that cannot be captured by simple Euclidean geometry. It would be important to extend the analysis to account for such manifolds, as this could provide a more accurate and robust understanding of how score and probability flows behave in high-dimensional, non-Euclidean spaces.

## Questions

(1) Can this study be expanded to the hypersphere of low dimension or a circle? Or a manifold with curvatures?

(2) If dropout was to be included to the model, do you think that there will be more virtual points generated?

(3) In the empirical results section, Figure (2) demonstrates the behavior of probability and score flows for the hyper-box dataset. Would it be possible to provide similar visualizations for the obtuse-angle and equilateral triangle datasets to further highlight the distinctions between these flows and how they behave on different data?

**Essential References Not Discussed:**

A tangential study of memorization in terms of the manifold --- [Ross et al. (2024)](https://arxiv.org/pdf/2411.00113)

A tangential study of the dynamical trajectory in diffusion models --- [Biroli et al. (2024)](https://www.nature.com/articles/s41467-024-54281-3)

A tangential study of generalization in diffusion models via the aspects of convolution ---
[Kamb and Ganguli (2025)](https://arxiv.org/pdf/2412.20292)

A tangential study of memorization and generalization in diffusion models via the lens of Hopfield models --- [Pham et al. (2024)](https://openreview.net/pdf?id=zVMMaVy2BY)

A tangential study of loss in manifold dimension which leads to generalization in diffusion models --- [Achilli et al. (2024)](https://arxiv.org/abs/2410.08727)

**Experimental Designs Or Analyses:**

I find no problems with the experimentation, including their designs and analyses. However, I do find the experimentation still somewhat lacking. Also, the experimentation detailed in Figures (12) and (13) feel redundant to me.

**Methods And Evaluation Criteria:**

The experiments used to confirm the theoretical claims are sound, and I do not see any problems with them. However, I would like to see experiments on manifold with curvatures.

**Other Comments Or Suggestions:**

N/A

**Other Strengths And Weaknesses:**

N/A

**Questions For Authors:**

See Above

**Relation To Broader Scientific Literature:**

This work is quite valuable, in my opinion. Firstly, it provides an analysis which differentiate PF and score flow ODEs, regarding their dynamical behavior. This distinction can be related to memorization and generalization (if done correctly). Specifically, the work provides some results on the fractions of sample types (e.g., virtual points and training points) generated from the diffusion model, regarding which flow is utilized.

Moreover, with these findings, I believe the authors can relate them to the aspects of Hopfield models and theory as well. Specifically, both virtual points and boundary points could be seen as spurious patterns, but as the training data size increases, most virtual points become generalized patterns. See [Pham et al. (2024)](https://openreview.net/pdf?id=zVMMaVy2BY).

**Theoretical Claims:**

Yes, I did check the theoretical claims, which I mentioned above, made by the paper.

Specifically, I spent a lot of time checking proofs behind Theorems (4.3 and 4.4). Such proofs are located in entirety of Appendix D.

Overall, I do not see any problems with the proofs in the main text and appendix as they were formatted and written decently.

---

> ### Author Rebuttal · Authors · 2025-04-01
>
> We greatly appreciate the reviewer’s kind words. We are glad to hear that the paper is presented in an accessible and engaging manner and that it was enjoyable to read.
> # Application to Higher Dimensional Data
> Indeed, we make simplifications to allow tractability. Please note, however, that while $N>D$  makes it impossible for the dataset to be exactly orthogonal, in the realistic regime $\exp(D)\gg N>D\gg 1$ most pairs are nearly orthogonal for standard Gaussian data. We believe this is a main reason why the orthogonality assumption is reasonable and common in high-dimensional theoretical analysis.
> Please see more details in our response to Reviewer SMjF under “Orthogonal Dataset Choice”.
> # Manifolds With Curvature
> We appreciate this insightful suggestion. Our theoretical analysis is based on Euclidean geometry and does not easily extend to nonlinear manifolds with curvature. Investigating how score and probability flows behave in such spaces is indeed an important direction for future research. Additionally, identifying meaningful experimental quantities analogous to those we defined (e.g., virtual points) in curved manifolds poses an interesting challenge.
> # Dropout Effect
> The impact of dropout on the generation of virtual points depends on several factors, including whether it is applied to the input, hidden layers, or both, as well as the dropout rate. Could the reviewer kindly elaborate on the specific dropout setting they have in mind?
> # Results for Additional Datasets
> We thank the reviewer for the suggestion. We will add additional visualizations to the appendix and highlight these distinctions in the revised version of the paper.
>
> # Connection to Hopfield Models
> We appreciate this insightful suggestion. In the revised version of the paper, we will incorporate the Memorization, Spurious, and Generalization metrics into our experimental results.
>
>
> # Essential References
> We thank the reviewer for these references; we will discuss these related works in the revised manuscript.

---

> > ### Comment · Reviewer_urF4 · 2025-04-01
> >
> > Dear authors,
> >
> > Thank you for your response. At first, I was a bit disappointed at your response to my comments since it was brief. But instead, I've taken a look at your response to other reviewers. Although I do not like looking responses to other reviewers since I'm afraid they might biased my opinion, I think your responses were good to me.
> >
> > Thus, I will raise my score from 3 to 4 and provide my comments and suggestions below.
> >
> > Firstly, I would like to justify my increase. I think this paper provides a valuable perspective on the trajectory behavior of score SDE and PF ODE, regarding on why one may work better than the other. Now, with the current results, I think the score of 3 is satisfactory for the paper since the story is still weak with them. **But I will take my chance with the authors and trust that they will add additional experiments, and make connections with Hopfield models regarding memorization, generalization, and possibly the emergence of spurious points as a way to contrast boundary and virtual points. I think this would be very interesting!**
> >
> > Regarding **dropout**, it would be interesting to see its effects applied on the hidden layers.
> >
> > Best of luck.

---

> > > ### Author Response · Authors · 2025-04-07
> > >
> > > We appreciate the reviewer’s trust and the thoughtful suggestions. In the short time we had, we were able to produce the following additional results.
> > >
> > > **Simulation Details**
> > >
> > > We computed the metrics from Pham et al. (2024) using the same setup as in Figure 2: the training set contains 31 orthogonal points, and we use the same 500 sampled points as the evaluation set (so $∣S∣=31$, $∣S^\text{eval}∣=500$ in the notation of Pham et al.). To construct the additional set $S^′$ such that $∣S^′∣=100\times∣S∣$, ensuring $S’$ is much larger than the training set (following the guidelines set by Pham et al.). Both $S^′$ and $S^\text{eval}$ were sampled using probability flow. We used the $L_{\infty}$​ metric as in Figure 2, and will include additional thresholds and distance metrics in the revised version.
> > >
> > > Our goal is to compare Pham et al.’s classification of evaluation points into *memorization*, *spurious*, and *generalization* categories with our own categorization into *training points*, *virtual points*, and *hyperbox boundary points*. We set $\delta_m​=0.2$, corresponding to the threshold used in our Figure 2 ($L_{\infty}​<0.2$). We used $\delta_s​=0.15$; additional values will be included in the revised paper.
> > >
> > > **Results:**
> > >
> > > * **Points marked as memorized** by Pham et al.:
> > >    * 100% are marked by us as training points
> > > * **Points marked as spurious:**
> > >    * 52.94% are marked by us as virtual points
> > >    * 0.98% are marked by us as training points
> > >    * 46.08% are marked by us as boundary points
> > > * **Points marked as a generalization:**
> > >    * 11.35% are marked by us as virtual points
> > >    * 4.86% are marked by us as training points
> > >    * 83.78% are marked by us as boundary points
> > >
> > > These results suggest that many virtual points are classified as *spurious* under the Pham et al. criteria.
> > > This matches our analysis: virtual points are stable, stationary points of the score flow; therefore, given a large evaluation set, we will have a cluster of points near the virtual points, which matches the spurious definition.
> > > However, due to the exponential number of virtual points, we cannot expect a cluster to form around each of them. As a result, some virtual points are also classified as “generalization”, under these metrics.

---

### Official Review · Reviewer_mtjQ · 2025-03-13

**Overall Recommendation:** 3

**Summary:**

The authors analyze the behaviour of score and probability flow ODEs when the score is estimated using min-cost shallow neural networks and restricted datasets. Under these assumptions, they derive theoretical results which find that the stationary points of PF-ODE and score flow ODE consist of summation of training set elements. Empirically, they confirm their theoretical findings by demonstrating the stability of the predicted stationary points and that PF-ODE and score flow ODE sampling converge to these samples.

## Update after Rebuttal

I'd like to thank the authors for performing additional experiments on my behalf. I sincerely appreciate the effort to produce these results in such a short amount of time.

I am fairly conflicted about this paper. I still am skeptical that these results are truly useful to the community. From the authors' latest reply, and simple linear algebra, for $N>=d$, any sample can be decomposed into a linear combination of the training set points so I struggle to determine what takeaways these finding have for practical diffusion applications. However, the theoretical results of this paper are sound, and are empirically well supported by the paper's experimental results. I have decided to update my score to a 3.

**Claims And Evidence:**

The only claim which I take issue with is the implied similarity between convergence to virtual and boundary points under these restrictive assumptions and the combination of semantic components of images as observed in Stable Diffusion outputs (see abstract, L236). This claim is made with little evidence, as to my understanding this work predicts convergence to hyperbox boundaries or vertices. It is not clear to me that "semantic sums" are equivalent to the dimension-wise sums of this work.

**Essential References Not Discussed:**

The related work presented in this manuscript is to the best of my knowledge not missing any essential references.

**Experimental Designs Or Analyses:**

A key assumption of this work is that the denoiser should fit the training data exactly. While this is completely reasonable for low noise levels where the modes of the dataset do not overlap, I am concerned about the training of the 100 other denoisers outside of the "low-noise regime". In this regime, a well trained denoiser should not exactly fit data points, but the posterior mean (ie eq. 2), which for a finite training set is a simple weighted average of the training points. A denoiser in this regime constrained to fit the training points exactly is therefore not a reasonable substitute for a well-trained diffusion model.

I am concerned that constrained optimization of the non "low-noise regime" denoisers may affect the conclusions reached from the empirical results. For example in Figure 6, with unconstrained optimization of the denoiser, no convergence to virtual points is observed in contrast to Figure 2 (b).

**Methods And Evaluation Criteria:**

The proposed evaluation methods are reasonable tests of their theoretical predictions

**Other Comments Or Suggestions:**

While first reading this work, I was confused by the term "scheduler" which is referenced in the abstract and introduction but is only defined on line 152. I suggest the authors introduce this concept earlier or perhaps replace it with "diffusion time scheduler" or similar to avoid confusion with other schedulers such as optimization schedules which can also have early stopping.

**Other Strengths And Weaknesses:**

Although I believe that the contributions of this paper are novel and generally well done, its main weakness is the strength of assumptions required to allow analytic tractability.

## Strengths

The problem of memorization in diffusion models is extremely relevant, and this paper's analysis of the probability flow ODE provides an interesting clue as to how memorization may occur in larger models. In addition, the presentation and writing are generally good. Finally, the empirical results generally confirm the predictions made by the theory and are sensible

## Weaknesses
The main weakness of this work are the restrictive assumptions. Specifically, the assumptions regarding orthogonality seem especially restrictive. In most deep learning settings, (ie stable diffusion) the number of data elements far exceeds the data dimensionality. The theoretical results also depend on a constrained optimization procedure which does not reflect the general practices for diffusion training. While I understand these assumptions are required for theoretical analysis, they also limit the contribution. Although the probability flow ODE results are potentially relevant to general diffusion sampling these results are questionable due to the optimization procedure of the denoiser.

**Questions For Authors:**

I've repeated the first two questions from prior sections of my review here for clarity

1. Regarding the min-cost solution of equation (17), why is the orthogonal assumption reasonable with Zeno et. al 2023 require obtuse angles between data points?
2. When training denoisers outside of the low-noise regime, why should denosiers be constrained to exactly fit the training data?
3. On line 123(R) you state "Notably, in contrast to the probability flow ODE, the min-cost denoiser here is independent of t" Could you clarify why the min-cost denoiser is dependent on the choice of ODE you wish to solve?
4. In your empirical results, you use 0.2 as a threshold for matching. Does this threshold have significance?
5. Across all of your experiments you find that samples can be categorized as Boundary, Training Pts. or Virtual Pts. Did you find any samples which fell outside of these classes?
6. In Figure 3 you evaluate the fraction of virtual points for $N \in [10, 15, 20, 25, 30\]$. How does the sample distribution change as you extend to $N > d$?

**Relation To Broader Scientific Literature:**

The contributions of this paper are somewhat complementary to the emerging literature on memorization in diffusion models, ie Somepalli et al. 2023 and Kadkhodie et al. 2024. However, the restrictive assumptions made in this paper make it somewhat less relevant to the general case of memorization/generalization in large scale diffusion models.

**Theoretical Claims:**

My main concern regarding theoretical claims is regarding equation (17) - the minimizer of the constrained optimization problem when trained on orthogonal data, ie $x_i^\top x_j = 0$. The authors cite Theorem 3 of Zeno et al. 2023 as the source of this minimzer. However, Theorem 3 of zeno et. has the condition of $x_i ^\top x_j < 0$. Could the authors please clarify why this relaxation is acceptable?

---

> ### Author Rebuttal · Authors · 2025-04-01
>
> We thank the reviewer for their thoughtful feedback.
> # Connection to Semantic Sums
> We are sorry for the lack of clarity. We did not aim to claim that the virtual points in orthogonal data must be equivalent to combinations of semantic components of images as observed in Stable Diffusion. This is only maintained as a motivation for virtual points. We agree that the current phrasing in the abstract may be confusing, and will change this in the revised paper.
> # Relaxation of Obtuse Angles
> The same arguments used in the proof of Theorem 3 in Zeno et al. (2023) can be applied to prove equation (17). The requirement for strictly obtuse angles (i.e., $x_i^Tx_j < 0$ instead of $x_i^Tx_j \leq 0$) in Zeno et al. (2023) is only made specifically to ensure the uniqueness of the solution. The proof of that theorem, therefore, applies to the orthogonal case as well, just without uniqueness. In other words, the function in (17) is a minimizer of (8), but not necessarily the unique minimizer.
>
> Specifically, the proof of Theorem 3 in Zeno et al. (2023) uses the fact that any unit active along two data points making an obtuse angle can be “split” into two units aligned with each data point with strictly lower representation cost. When the rays are orthogonal, the two “split” units may have the same representation cost as the original unit. So while the interpolant given by equation (17) is still a minimizer, it is no longer necessarily unique in the case of orthogonal data.
>
> We will clarify this in the paper.
>
> # Constraining Denoisers to Exactly Fit the Training Data Outside the Low-Noise Regime
> Indeed, outside of the low-noise regime, the denoisers would not exactly fit the training data, and the min-norm denoiser in this case is not defined (as we can not get zero loss).
> We use the same training regime for all denoisers for consistency. To complement that, we conduct additional simulations in App. E, where we train the neural denoisers using a standard training procedure using the Adam optimizer with weight decay, obtaining similar results. We hypothesize that this is because high-noise denoisers primarily influence the initial condition, before the critical noise threshold is crossed, and we start the “useful” final sampling phase [R1].
> # Strength of Assumptions
> Indeed, we make simplifications to allow tractability. Please note, however, that while $N>D$  makes it impossible for the dataset to be exactly orthogonal, in the realistic regime $\exp(D) \gg N>D\gg 1$ most pairs are nearly orthogonal for standard Gaussian data. We believe this is a main reason why the orthogonality assumption is reasonable and common in high-dimensional theoretical analysis.
> Please see more details in our response to Reviewer SMjF under “Orthogonal Dataset Choice”.
> # The Constrained Optimization Procedure
> App. E holds additional simulations where we train the neural denoisers using a standard training procedure with the Adam optimizer, with or without weight decay regularization (which promotes min-norm results, but not forces it). Specifically, for weight decay, we tried $\lambda=0.25,0.5,1$. All values led to similar results, therefore we included in Fig. 6 only the case of $\lambda=0.25$. As can be seen, in the case of WD=0 we converge only to the training points or boundary points of the hyperbox, whereas for WD>0 we converge to virtual points as well, which aligns with the results achieved with the Augmented Lagrangian method.
> # Diffusion Time Scheduler
> Thanks for this suggestion, we will change the terminology in the revised paper.
> # Min-cost Denoiser Independent of $t$
> The time dependence of the min-cost denoiser is through $\rho$. Specifically, in eq.11 (probability flow ODE), the RHS includes a derivative w.r.t  $t$ of $\sigma_t^2$ times the score function, so by applying time re-scaling arguments, we get a time-dependent $\rho$, and, therefore, denoiser. In eq. 13, however, the RHS does not include a derivative with respect to $t$, so $\rho$ and the denoiser are time-independent.
> # Significance of the 0.2 Threshold
> We show the effect of changing the thresholds values in App. F. Qualitatively, this does not drastically affect the results.
> # Samples Beyond Training/Virtual/Boundary Points
> No, all points converged to one of the 3 categories, which matches our theoretical analysis.
> # Extending Fig. 3 to $N>d$
> We thank the reviewer for this suggestion. In strictly orthogonal data, it is impossible to have $N>d$. If the reviewer has a different dataset in mind, we will be happy to train all our denoisers (150) on the relevant $N$ and $d$ on this new dataset, and include our findings in the revised paper. Note also that if we change the dataset, it can become challenging to define and find the virtual points.
>
> [R1] Gabriel Raya & Luca Ambrogioni. Spontaneous symmetry breaking in generative diffusion models. NeurIPS, 2023.

---

> > ### Comment · Reviewer_mtjQ · 2025-04-02
> >
> > Thank you to the reviewers for their response to my initial review. Most of the items I raised have been reasonably addressed by your rebuttal. I am leaning towards increasing my score on this basis.
> >
> > My only remaining item of concern is the applicability of your work to larger, non-orthogonal datasets. I have read your response to SMjF. I am satisfied that most pairs of data in this regime are nearly orthogonal and that this assumption is not as restrictive as I feared. However, I am still unsure of how to reconcile this near-orthogonality with your response to my question regarding Fig 3. for $N > D$. If you cannot define virtual points in this regime, I am concerned that the findings of this work are not useful for understanding generalization for general datasets. I would appreciate if the authors could expand upon how their findings can be understood in this setting.
> >
> > In response to your request for a dataset, I would suggest perhaps augmenting your original orthonormal training set with additional samples randomly drawn from the surface of the hyperbox. Do you still find that training points exclusively converge to vertices and edges of the box in this case?

---

> > > ### Author Response · Authors · 2025-04-07
> > >
> > > **Defining Virtual Points**
> > >
> > > We apologize for not being sufficiently clear. Virtual points are always easy to define conceptually, as well as for theoretical analysis, as "all points which are sums of clean training points".
> > > The problem lies in how to define a point 'x' as "virtual" when examining the result of a numerical simulation, for non-orthogonal data. For example, if $N \gg d$ and the training points have acute angles, then to find if 'x' is a 'k'-order virtual point, we need to search over k-choose-N combinations of the training points, which can quickly become a prohibitively large number as k and N increase. For each combination, we need to decide if the point 'x' can be decomposed as a sum of these training points, up to some tolerances (which are required since we have a noisy simulation). Choosing good values for the tolerances is an interesting statistical problem when the angles are acute. Lastly, it is harder to visualize the virtual point in this case (we can't just project to the hypercube like we did here).
> > >
> > > **New simulation Details**
> > >
> > > Following the reviewer’s suggestion, we augmented the original orthonormal dataset used in Figure 2 (which included 31 orthogonal data points in D=30) with additional random data points generated as follows:
> > > * We sampled a random vector with i.i.d. elements from the uniform distribution [0.3,0.7]. This choice avoids the degenerate case where no denoisers are active in the low-noise regime.
> > > * We then projected the vector on a random face of the hyperbox to ensure that the new random data points lie on the hyperbox surface.
> > >
> > > We trained the same neural denoisers in Figure 2, using M=500 noisy samples. We considered two cases: N=40 and N=50. As in the original experiment, we used AL optimization with $L_{\infty}$ metric with 0.2 threshold.
> > >
> > > **Simulation Results**
> > >
> > > As can be seen from the results below, in these cases (when N>D), the probability flow almost exclusively converges to the hyperbox surface: either to the boundary, training points (either old orthogonal points or the new points on the surface), or to other vertices of the hyperbox (the original virtual points).
> > >
> > > ||$N=40$|$N=50$|
> > > |----|----|----|
> > > |hyperbox surface|99%|98.21%|
> > > |original virtual points|2.4%|3.4%|
> > > |orthogonal training datapoints|32%|19.6%|
> > > |new random data points|14.6%|19%|
> > >
> > > We will add to the revised paper the results for different metrics and thresholds.

---

### Official Review · Reviewer_SMjF · 2025-03-13

**Overall Recommendation:** 4

**Summary:**

The authors analyze memoization of diffusion models in shallow relu networks. To analyze this, they consider the probability flow ode, and obtain additional results by introducing a simpler score flow. Under some assumptions, they show that both ODE's have stationary points corresponding to the training points, and in some cases sums thereof. They verify their theoretical claims through various experiments on small orthogonal datasets and confirm their theory holds

## update after review
The authors addressed my concern about N > D well, and I have increased my score from 3 to 4 as a result

**Claims And Evidence:**

It seems to me the claims match the evidence.

**Essential References Not Discussed:**

NA

**Experimental Designs Or Analyses:**

I think the choice of orthogonal dataset is interesting, but potentially understandable, please see my question under "strenghts and weaknesses"

**Methods And Evaluation Criteria:**

The evaluation is on simple datasets, but seems appropriate for the problem setting, i.e. a theoretical analysis of neural networks for diffusion modeling

**Other Comments Or Suggestions:**

NA

**Other Strengths And Weaknesses:**

Overall the paper is well written and good to follow, even for someone with limited knowledge of theoretical analysis of shallow neural networks. The theorems and proofs are well explained, and seem (to the best of my ability of judging) to be correct. While the problem set up is simple, it is understandable to the extent that a rigorous theoretical analysis of anything more complicated quickly becomes intractable, I do have some questions:
- The authors state that a standard normal distribution becomes more orthogonal under increasing dimension. While this is true, most machine learning datasets have N >> D, and can therefore not be orthogonal. Can the authors defend their choice of orthogonal dataset and why this would be relevant to actual datasets?
- Can the authors explain what they mean by "interpolates the training data"? Do the authors mean that in an open ball around a training point the denoiser maps to the training data? This seems to be what the definitions say but this is not what I would consider "interpolation".
- Can the authors elaborate whether virtual points correspond to generalization? If I understand the argument correctly, the authors interpret the training-data "copy-pasting" sometimes observed in diffusion models that way. Is it correct to understand this "copy-pasting" as the virtual training points discussed in the theory? Ultimately: are virtual points desirable, or not?
- Following up on this question, for an orthogonal dataset, what does it mean to "generalize"?  Or perhaps a better question is, is the underlying distribution simply a categorical distribution?
- Is it possible to model the noise distribution as a true noise distribution rather than a fixed collection of points? Why did the authors choose to use a fixed set of points for the noise distribution?

**Questions For Authors:**

See strengths and weaknesses

**Relation To Broader Scientific Literature:**

-

**Theoretical Claims:**

The theorems are well explained, and seem (to the best of my ability of judging) to be correct. I checked the correctness of the proofs in Section 3 in appendix A.

---

> ### Author Rebuttal · Authors · 2025-04-01
>
> We sincerely appreciate the reviewer’s positive feedback.
> # Orthogonal Dataset Choice
> First, when $N>D$, it is indeed impossible for the dataset to be exactly orthogonal. However, for standard i.i.d. Gaussian data $x_n$, it is easy to show  (e.g., using the analysis in Section 3.2.3 of [R1] and the union bound) that the largest cosine of the angle between two different datapoints is, with high probability,
> $\max_{n\neq m} \frac{x_n \cdot x_m}{\|x_n\| \|x_m\|} \sim \sqrt{\frac{\ln N}{d}}$.
> Therefore, in the realistic regime $\exp(D) \gg N>D \gg 1$, most pairs are nearly orthogonal. We believe this is a main reason why the orthogonality assumption is reasonable and common (e.g., [R2]) in high-dimensional theoretical analysis.
>
> Second, to make sure we are on the same page, please note that our analysis goes beyond the orthogonal case and covers all the analytically solvable cases found in Zeno et al. 2023, though some of this analysis was relegated to the appendix. For example, in App. D (line 630), we extend our analysis beyond the strictly orthogonal case by considering the scenario where the convex hull of the training points forms an obtuse-angle simplex. We showed there that similar behavior emerges even when strict orthogonality does not hold, supporting the broader relevance of our findings. In the main paper, we indeed focused on orthogonal datasets: this was done to simplify the presentation, and since the solution is qualitatively similar to that in the obtuse angle case.
> # Interpolation Meaning
> Yes, by "interpolates the training data," we mean that the denoiser maps each noisy sample to its corresponding clean training point, which leads to a zero empirical loss. Solutions with zero empirical loss are commonly called interpolators in the ML theoretical literature (e.g.,
> [R3]). As an approximation to this, we further assume that the denoiser maps the entire open ball centered around each clean data point to that clean point (this was also done and explained in Zeno et al.).
> # Virtual Points & Generalisation
> Yes, we view these virtual points as desirable, since they go beyond the empirical data distribution, and create new combinations not seen before in the training data. It is perhaps more accurate to view this on some scale: low-order combinations (e.g., which can be easily observed as “copy-paste”) may seem closer to memorization (“low creativity”), while high-order combinations show a higher degree of generalization (“high creativity”). We will discuss this interesting point in the paper.
> # Generalisation in Orthogonal Datasets
> In the case of an orthogonal dataset, we interpret the hyperbox boundary as an implicit data manifold, even though we do not assume an explicit sampling model that generates the training data (e.g., a distribution supported on the manifold). In this context, we define generalization as the ability to sample points from the hyperbox boundary.
> # Noise Distribution Modeling
> In practice, diffusion models are trained for a finite number of iterations using Gaussian noise. Consequently, during training, we effectively observe (different) noisy samples that all lie within an open ball around each clean data point (see additional details in sec. 3 in Zeno et al. 2023).
> In the multivariate case, finding an exact min-cost solution for finitely many noise realizations is generally intractable. So, we assume that the denoiser maps each open ball to its corresponding clean data point.
>
> [R1] Roman Vershynin. High-Dimensional Probability. 2019.
>
> [R2] Andrew M. Saxe et al. Exact solutions to the nonlinear dynamics of learning in deep linear neural networks. arXiv:1312.6120, 2013.
>
> [R3] Siyuan Ma et al. The Power of Interpolation: Understanding the Effectiveness of SGD in Modern Over-parametrized Learning. ICML, 2018.

---

### Decision · Program_Chairs · 2025-05-01

**Decision:**

Accept (poster)

**Comment:**

This paper was reviewed by the four experts in the field. The reviewers agreed that the paper has some important theoretical insights explaning mechanisms of how score and probability flow samplings work. While the problem setup is simple, more complicated theoretical setups could be less tractable. In particular, the authors proved that
(1) score and probability flow samplings follow similar trajectories in case of orthogonal data,
(2) probability flow can generate more non-trivial points on a manifold: this sampling can generate both training points and new points being sums of the training points.

The paper proposes some theoretical insigts on mechanisms of memorization in diffusion models, although due to restrictive assumptions used in this paper the conclusions are less relevant to the general case of memorization/generalization in large scale diffusion models.

The reviewers did raise some valuable concerns that should be addressed in the final camera-ready version of the paper. The authors are encouraged to make the necessary changes to the best of their ability. All additional simulation experiments, provided by the authors during the rebuttal phase, should be included.

Also, in the revised final version of the paper it is important to discuss
- orthogonal dataset choice and interpolation meaning (reviewers SMjF, mtjQ)
- Relaxation of Obtuse Angles, possibly including the theoretical derivation similar to the derivation from Zeno et al. (2023) (reviewer mtjQ)
- dropout effect on the hidden layers and Connection to Hopfield Models (reviewer urF4)
- Implications for Practical Image Diffusion Models (reviewer vy6L). I would propose to add some experiments confirming the claims, stated in the authors’ rebuttal to this reviewer.

 We congratulate the authors on the acceptance of their paper!